# A Computationally Efficient Method for Learning Exponential Family Distributions

**Abhin Shah**
MIT
abhin@mit.edu

**Devavrat Shah**
MIT
devavrat@mit.edu

**Gregory W. Wornell**
MIT
gww@mit.edu

## Abstract

We consider the question of learning the natural parameters of a $k$-parameter *minimal* exponential family from i.i.d. samples in a computationally and statistically efficient manner. We focus on the setting where the support as well as the natural parameters are appropriately bounded. While the traditional maximum likelihood estimator for this class of exponential family is consistent, asymptotically normal, and asymptotically efficient, evaluating it is computationally hard. In this work, we propose a computationally efficient estimator that is consistent as well as asymptotically normal under mild conditions. We provide finite sample guarantees to achieve an ($\ell_2$) error of $\alpha$ in the parameter estimation with sample complexity $O(\text{poly}(k/\alpha))$ and computational complexity $O(\text{poly}(k/\alpha))$. To establish these results, we show that, at the population level, our method can be viewed as the maximum likelihood estimation of a re-parameterized distribution belonging to the same class of exponential family.

## 1 Introduction

We are interested in the problem of learning the natural parameters of a *minimal* exponential family with bounded support. Consider a $p$-dimensional random vector $\mathbf{x} = (x_1, \cdots, x_p)$ with support $\mathcal{X} \subset \mathbb{R}^p$. An exponential family is a set of parametric probability distributions with probability densities of the following canonical form

$$f_{\mathbf{x}}(\mathbf{x}; \boldsymbol{\theta}) \propto \exp\left(\boldsymbol{\theta}^T \boldsymbol{\phi}(\mathbf{x}) + \beta(\mathbf{x})\right), \tag{1}$$

where $\mathbf{x} \in \mathcal{X}$ is a realization of the underlying random variable $\mathbf{x}$, $\boldsymbol{\theta} \in \mathbb{R}^k$ is the natural parameter, $\boldsymbol{\phi} : \mathcal{X} \to \mathbb{R}^k$ is the natural statistic, $k$ denotes the number of parameters, and $\beta$ is the log base function. For representational convenience, we shall utilize the following equivalent representation of (1):

$$f_{\mathbf{x}}(\mathbf{x}; \Theta) \propto \exp\left(\left\langle\left\langle \Theta, \Phi(\mathbf{x}) \right\rangle\right\rangle\right) = \exp\left(\sum_{i \in [k_1], j \in [k_2], l \in [k_3]} \Theta_{ijl} \times \Phi_{ijl}(\mathbf{x})\right) \tag{2}$$

where $\Theta = [\Theta_{ijl}] \in \mathbb{R}^{k_1 \times k_2 \times k_3}$ is the natural parameter, $\Phi = [\Phi_{ijl}] : \mathcal{X} \to \mathbb{R}^{k_1 \times k_2 \times k_3}$ is the natural statistic, $k_1 \times k_2 \times k_3 - 1 = k$, and $\left\langle\left\langle \Theta, \Phi(\mathbf{x}) \right\rangle\right\rangle$ denotes the tensor inner product, i.e., the sum of product of entries of $\Theta$ and $\Phi(\mathbf{x})$. An exponential family is *minimal* if there does not exist a nonzero tensor $\mathbf{U} \in \mathbb{R}^{k_1 \times k_2 \times k_3}$ such that $\left\langle\left\langle \mathbf{U}, \Phi(\mathbf{x}) \right\rangle\right\rangle$ is equal to a constant for all $\mathbf{x} \in \mathcal{X}$.

The notion of exponential family was first introduced by Fisher [17] and was later generalized by Darmois [12], Koopman [30], and Pitman [40]. Exponential families play an important role in statistical inference and arise in many diverse applications for a variety of reasons: (a) they are analytically tractable, (b) they arise as the solutions to several natural optimization problems on the space of probability distributions, (c) they have robust generalization property (see [5, 2] for details).

*Truncated* (or bounded) exponential family, first introduced by Hogg and Craig [20], is a set of parametric probability distributions resulting from truncating the support of an exponential family. *Truncated* exponential families share the same parametric form with their non-truncated counterparts up to a normalizing constant. These distributions arise in many applications where we can observe only a truncated dataset (truncation is often imposed by during data acquisition) e.g., geolocation tracking data can only be observed up to the coverage of mobile signal, police department can often monitor crimes only within their city's boundary.

The natural parameter $\Theta$ specifies a particular distribution in the exponential family. If the natural statistic $\Phi$ and the support of $\mathbf{x}$ (i.e., $\mathcal{X}$) are known, then learning a distribution in the exponential family is equivalent to learning the corresponding natural parameter $\Theta$. Despite having a long history, there has been limited progress on learning natural parameter $\Theta$ of a *minimal truncated* exponential family. More precisely, there is no known method (without any abstract condition) that is both computationally and statistically efficient for learning natural parameter of the *minimal truncated* exponential family considered in this work.

## 1.1  Contributions

As the primary contribution of this work, we provide a computationally tractable method with statistical guarantees for learning distributions in *truncated minimal* exponential families. Formally, the learning task of interest is estimating the true natural parameter $\Theta^*$ from i.i.d. samples of $\mathbf{x}$ obtained from $f_{\mathbf{x}}(\cdot; \Theta^*)$. We focus on the setting where $\Theta^*$ and $\Phi$ are appropriately bounded (see Section 2). We summarize our contributions in the following two categories.

**1. Computationally Tractable Estimator : Consistency, Normality, Finite Sample Guarantees.** Given $n$ samples $\mathbf{x}^{(1)} \cdots, \mathbf{x}^{(n)}$ of $\mathbf{x}$, we propose the following novel loss function to learn a distribution belonging to the exponential family in (2):

$$\mathcal{L}_n(\Theta) = \frac{1}{n} \sum_{t=1}^{n} \exp\big( - \langle\langle \Theta, \varPhi(\mathbf{x}^{(t)}) \rangle\rangle \big), \tag{3}$$

where $\varPhi(\cdot) = \Phi(\cdot) - \mathbb{E}_{\mathcal{U}_{\mathcal{X}}}[\Phi(\cdot)]$ with $\mathcal{U}_{\mathcal{X}}$ being the uniform distribution over $\mathcal{X}$. We establish that the estimator $\hat{\Theta}_n$ obtained by minimizing $\mathcal{L}_n(\Theta)$ over all $\Theta$ in the constraint set $\Lambda$, i.e.,

$$\hat{\Theta}_n \in \operatorname*{arg\,min}_{\Theta \in \Lambda} \mathcal{L}_n(\Theta), \tag{4}$$

is consistent and (under mild further restrictions) asymptotically normal (see Theorem 4.2). We obtain an $\epsilon$-optimal solution $\hat{\Theta}_{\epsilon,n}$ of the convex minimization problem in (4) (i.e., $\mathcal{L}_n(\hat{\Theta}_{\epsilon,n}) \leq \mathcal{L}_n(\hat{\Theta}_n) + \epsilon$) by implementing a projected gradient descent algorithm with $O(\text{poly}(k_1 k_2/\epsilon))^1$ iterations (see Lemma 3.1). Finally, we provide rigorous finite sample guarantees for $\hat{\Theta}_{\epsilon,n}$ (with $\epsilon = O(\alpha^2)$) to achieve an error of $\alpha$ (in the tensor $\ell_2$ norm) with respect to the true natural parameter $\Theta^*$ with $O(\text{poly}(k_1 k_2/\alpha))$ samples and $O(\text{poly}(k_1 k_2/\alpha))$ computations (see Theorem 4.3). By letting certain additional structure on the natural parameter, we allow our framework to capture various constraints on the natural parameter including sparse, low-rank, sparse-plus-low-rank (see Section 2.1).

**2. Connections to maximum likelihood estimation (MLE) of a re-parameterized distribution.** We establish connections between our method and the MLE of the distribution $f_{\mathbf{x}}(\cdot; \Theta^* - \Theta)$. We show that the estimator that minimizes the population version of the loss function in (3) i.e.,

$$\mathcal{L}(\Theta) = \mathbb{E}\Big[ \exp\big( - \langle\langle \Theta, \varPhi(\mathbf{x}) \rangle\rangle \big) \Big].$$

is equivalent to the estimator that minimizes the Kullback-Leibler (KL) divergence between $\mathcal{U}_{\mathcal{X}}$ (the uniform distribution on $\mathcal{X}$) and $f_{\mathbf{x}}(\cdot; \Theta^* - \Theta)$ (see Theorem 4.1). Therefore, at the population level, our method can be viewed as the MLE of the parametric family $f_{\mathbf{x}}(\cdot; \Theta^* - \Theta)$. We show that the KL divergence (and therefore $\mathcal{L}(\Theta)$) is minimized if and only if $\Theta = \Theta^*$, and this connection provides an intuitively pleasing justification of the estimator in (4).

---

[1]We let $k_3 = O(1)$. See Section 2.

## 1.2 Related Works

In this section, we look at the related works on learning exponential family. Broadly speaking, there are two line of approaches to overcome the computational hardness of the MLE : (a) approximating the MLE and (b) selecting a surrogate objective. Given the richness of both of approaches, we cannot do justice in providing a full overview. Instead, we look at a few examples from both. Next, we look at some of the related works that focus on learning a class of exponential family. More specifically, we look at works on (a) learning the Gaussian distribution and (b) learning exponential family Markov random fields (MRFs). Finally, we explore some works on the powerful technique of score matching. In Appendix A, we further review works on learning exponential family MRFs, score-based methods (including the related literature on Stein discrepancy) and latent variable graphical models (since these capture sparse-plus-low-rank constraints on the parameters similar to our framework).

**Approximating the MLE.** Most of the techniques falling in this category approximate the MLE by approximating the log-partition function. A few examples include : (a) approximating the gradient of log-likelihood with a stochastic estimator by minimizing the contrastive divergence [19]; (b) upper bounding the log-partition function by an iterative tree-reweighted belief propagation algorithm [57]; (c) using Monte Carlo methods like importance sampling for estimating the partition function [43]. Since these methods approximate the partition function, they come at the cost of an approximation error or result in a biased estimator.

**Selecting surrogate objective.** This line of approach selects an easier-to-compute surrogate objective that completely avoids the partition function. A few examples are as follows : (a) pseudo-likelihood estimators [4] approximate the joint distribution with the product of conditional distributions, each of which only represents the distribution of a single variable conditioned on the remaining variables; (b) score matching [22, 21] minimizes the Fisher divergence between the true log density and the model log density. Even though score matching does not require evaluating the partition function, it is computationally expensive as it requires computing third order derivatives for optimization; (c) kernel Stein discrepancy [32, 9] measures the kernel mean discrepancy between a data distribution and a model density using the Stein's identity. This measure is directly characterized by the choice of the kernel and there is no clear objective for choosing the right kernel [61].

**Learning the Gaussian distribution.** Learning the Gaussian distribution is a special case of learning exponential family distributions. There has been a long history of learning Gaussian distributions in the form of learning Gaussian graphical models e.g. the neighborhood selection scheme [36], the graphical lasso [18], the CLIME [6], etc. However, finite sample analysis of these methods require various hard-to-verify conditions e.g. the restricted eigenvalue condition, the incoherence assumption ([59, 24]), bounded eigenvalues of the precision matrix, etc. A recent work [28] provided an algorithm whose sample complexity, for a specific subclass of Gaussian graphical models, match the information-theoretic lower bound of [60] without the aforementioned hard-to-verify conditions.

**Learning Exponential Family Markov Random Fields (MRFs).** MRFs can be naturally represented as exponential family distributions via the principle of maximum entropy (see [58]). A popular method for learning MRFs is estimating node-neighborhoods (fitting conditional distributions of each node conditioned on the rest of the nodes) because the natural parameter is assumed to be node-wise-sparse. A recent line of work has considered a subclass of node-wise-sparse pairwise continuous MRFs where the node-conditional distribution of $x_i \in \mathcal{X}_i$ for every $i$ arise from an exponential family as follows:

$$f_{\mathsf{x}_i|\mathsf{x}_{-i}}(x_i|\mathbf{x}_{-i} = x_{-i}) \propto \exp\left(\left[\theta_i + \sum_{j \in [p], j \neq i} \theta_{ij}\phi(x_j)\right]\phi(x_i)\right), \tag{5}$$

where $\phi(x_i)$ is the natural statistics and $\theta_i + \sum_{j \in [p], j \neq i} \theta_{ij}\phi(x_j)$ is the natural parameter.[2] Yang et al. [62] showed that only the following joint distribution is consistent with the node-conditional distributions in (5) :

$$f_{\mathbf{x}}(\mathbf{x}) \propto \exp\left(\sum_{i \in [p]} \theta_i\phi(x_i) + \sum_{j \neq i} \theta_{ij}\phi(x_i)\phi(x_j)\right). \tag{6}$$

To learn the node-conditional distribution in (5) for linear $\phi(\cdot)$ (i.e., $\phi(x) = x$), Yang et al. [62] proposed an $\ell_1$ regularized node-conditional log-likelihood. However, their finite sample analysis

---

[2]Under node-wise-sparsity, $\sum_{j \in [p], j \neq i} |\theta_{ij}|$ is bounded by a constant for every $i \in [p]$.

required the following conditions: incoherence, dependency (see [59, 24]), bounded moments of the variables, and local smoothness of the log-partition function. Tansey et al. [51] extended the approach in [62] to vector-space MRFs (i.e., vector natural parameters and natural statistics) and non-linear $\phi(\cdot)$. They proposed a sparse group lasso (see [45]) regularized node-conditional log-likelihood and an alternating direction method of multipliers based approach to solving the resulting optimization problem. However, their analysis required same conditions as [62].

While node-conditional log-likelihood has been a natural choice for learning exponential family MRFs, M-estimation [56, 55, 44] and maximum pseudo-likelihood estimator [39, 63, 10] have recently gained popularity. The objective function in M-estimation is a sample average and the estimator is generally consistent and asymptotically normal. Shah et al. [44] proposed the following M-estimation (inspired from [56, 55]) for vector-space MRFs and non-linear $\phi(\cdot)$: with $\mathcal{U}_{\mathcal{X}_i}$ being the uniform distribution on $\mathcal{X}_i$ and $\tilde{\phi}(x_i) = \phi(x_i) - \int_{x_i'} \phi(x_i')\mathcal{U}_{\mathcal{X}_i}(x_i')dx_i'$

$$\arg\min \frac{1}{n} \sum_{i=1}^{n} \exp\left(-\left[\theta_i \tilde{\phi}(x_i) + \sum_{j \in [p], j \neq i} \theta_{ij} \tilde{\phi}(x_i)\tilde{\phi}(x_j)\right]\right). \tag{7}$$

They provided an entropic descent algorithm (borrowing from [55]) to solve the optimization in (7) and their finite-sample bounds rely on bounded domain of the variables and a condition (naturally satisfied by linear $\phi(\cdot)$) that lower bounds the variance of a non-constant random variable.

Yuan et al. [64] considered a broader class of sparse pairwise exponential family MRFs compared to [62]. They studied the following joint distribution with natural statistics $\phi(\cdot)$ and $\psi(\cdot)$

$$f_{\mathbf{x}}(\mathbf{x}) \propto \exp\left(\sum_{i \in [p]} \theta_i \phi(x_i) + \sum_{j \neq i} \theta_{ij} \psi(x_i, x_j)\right). \tag{8}$$

They proposed an $\ell_{2,1}$ regularized joint likelihood and an $\ell_{2,1}$ regularized node-conditional likelihood. They also presented a Monte-Carlo approximation to these estimators via proximal gradient descent. Their finite-sample analysis required restricted strong convexity (of the Hessian of the negative log-likelihood of the joint density) and bounded moment-generating function of the variables.

Building upon [55] and [44], Ren et al. [41] addressed learning continuous exponential family distributions through a series of numerical experiments. They considered unbounded distributions and allowed for terms corresponding to multi-wise interactions in the joint density. However, they considered only monomial natural statistics. Further, they assume node-wise-sparsity of the parameters as in MRFs and their estimator is defined as a series of node-wise optimization problems.

In summary, tremendous progress has been made on learning the sub-classes of exponential family in (6) and (8). However, this sub-classes are restricted by the assumption that the natural parameters are node-wise-sparse. For example, none of the existing methods for exponential family MRFs work in the setting where the natural parameters have a low-rank constraint.

**Score-based method.** A scoring rule $S(\mathbf{x}, Q)$ is a numerical score assigned to a realization $\mathbf{x}$ of a random variable $\mathbf{x}$ and it measures the quality of a predictive distribution $Q$ (with probability density $q(\cdot)$). If $P$ is the true distribution of $\mathbf{x}$, the divergence $D(P, Q)$ associated with a scoring rule is defined as $\mathbb{E}_P[S(\mathbf{x}, Q) - S(\mathbf{x}, P)]$. The MLE is an example of a scoring rule with $S(\cdot, Q) = -\log q(\cdot)$ and the resulting divergence is the KL-divergence.

To bypass the intractability of MLE, [22] proposed an alternative scoring rule with $S(\cdot, Q) = \Delta \log q(\cdot) + \frac{1}{2}\|\nabla \log q(\cdot)\|_2^2$ where $\Delta$ is the Laplacian operator, $\nabla$ is the gradient and $\|\cdot\|_2$ is the $\ell_2$ norm. This method is called *score matching* and the resulting divergence is the Fisher divergence. Score matching is widely used for estimating unnormalizable probability distributions because computing the scoring rule $S(\cdot, Q)$ does not require knowing the partition function. Despite the flexibility of this approach, it is computationally expensive in high dimensions since it requires computing the trace of the unnormalized density's Hessian (and its derivatives for optimization). Additionally, it breaks down for models in which the second derivative grows very rapidly.

In [34], the authors considered estimating truncated exponential family using the principle of score matching. They build on the framework of generalized score matching [21] and proposed a novel estimator that minimizes a weighted Fisher divergence. They showed that their estimator is a special case of minimizing a Stein Discrepancy. However, their finite sample analysis relies on certain hard-to-verify assumptions, for example, the assumption that the optimal parameter is well-separated

from other neighboring parameters in terms of their population objective. Further, their estimator lacks the useful properties of asymptotic normality and asymptotic efficiency.

## 1.3 Useful notations and outline

**Notations.** For any positive integer $t$, let $[t] := \{1, \cdots, t\}$. For a deterministic sequence $v_1, \cdots, v_t$, we let $\mathbf{v} := (v_1, \cdots, v_t)$. For a random sequence $\mathsf{v}_1, \cdots, \mathsf{v}_t$, we let $\mathbf{v} := (\mathsf{v}_1, \cdots, \mathsf{v}_t)$. For a matrix $\mathbf{M} \in \mathbb{R}^{u \times v}$, we denote the element in $i^{th}$ row and $j^{th}$ column by $M_{ij}$, the singular values of the matrix by $\sigma_i(\mathbf{M})$ for $i \in [\min\{u, v\}]$, the matrix maximum norm by $\|\mathbf{M}\|_{\max} := \max_{i \in [u], j \in [v]} |M_{ij}|$, the entry-wise $L_{1,1}$ norm by $\|\mathbf{M}\|_{1,1} := \sum_{i \in [u], j \in [v]} |M_{ij}|$, the nuclear norm by $\|\mathbf{M}\|_\star := \sum_{i \in [\min\{u,v\}]} \sigma_i(\mathbf{M})$. We denote the Frobenius or Trace inner product of matrices $\mathbf{M}, \mathbf{N} \in \mathbb{R}^{u \times v}$ by $\langle \mathbf{M}, \mathbf{N} \rangle := \sum_{i \in [u], j \in [v]} M_{ij} N_{ij}$. For a matrix $\mathbf{M} \in \mathbb{R}^{u \times v}$, we denote a generic norm on $\mathbb{R}^{u \times v}$ by $\mathcal{R}(\mathbf{M})$ and denote the associated dual norm by $\mathcal{R}^*(\mathbf{M}) := \sup\{\langle \mathbf{M}, \mathbf{N} \rangle | \mathcal{R}(\mathbf{N}) \le 1\}$ where $\mathbf{N} \in \mathbb{R}^{u \times v}$. For a tensor $\mathbf{U} \in \mathbb{R}^{u \times v \times w}$, we denote its $(i, j, l)$ entry by $U_{ijl}$, its $l^{th}$ slice (obtained by fixing the last index) by $U_{::l}$ or $U^{(l)}$, the tensor maximum norm (with a slight abuse of notation) by $\|\mathbf{U}\|_{\max} := \max_{i \in [u], j \in [v], l \in [w]} |U_{ijl}|$, and the tensor norm by $\|\mathbf{U}\|_{\mathrm{T}} := \sqrt{\sum_{i \in [u], j \in [v], l \in [w]} U_{ijl}^2}$. We denote the tensor inner product of tensors $\mathbf{U}, \mathbf{V} \in \mathbb{R}^{u \times v \times w}$ by $\langle\langle \mathbf{U}, \mathbf{V} \rangle\rangle := \sum_{i \in [u], j \in [v], l \in [w]} U_{ijl} V_{ijl}$. We denote the vectorization of the tensor $\mathbf{U} \in \mathbb{R}^{u \times v \times w}$ by $\mathrm{vec}(\mathbf{U}) \in \mathbb{R}^{uvw \times 1}$ (the ordering of the elements is not important as long as it is consistent). Let $\mathbf{0} \in \mathbb{R}^{k_1 \times k_2 \times k_3}$ denote the tensor with every entry zero. We denote a $p$-dimensional ball of radius $b$ centered at $0$ by $\mathcal{B}(0, b)$.

**Outline.** In Section 2, we formulate the problem of interest, state our assumptions, and provide examples. In Section 3, we provide our loss function and algorithm. In Section 4, we present our main results including the connections to the MLE of $f_{\mathbf{x}}(\cdot; \Theta^* - \Theta)$, consistency, asymptotic normality, and finite sample guarantees. In Section 5, we conclude, provide some remarks, discuss limitations as well as some directions for future work. See supplementary for organization of the Appendix.

## 2 Problem Formulation

Let $\mathbf{x} = (\mathsf{x}_1, \cdots, \mathsf{x}_p)$ be a $p-$dimensional vector of continuous random variables.[3] For any $i \in [p]$, let the support of $\mathsf{x}_i$ be $\mathcal{X}_i \subset \mathbb{R}$. Define $\mathcal{X} := \prod_{i=1}^p \mathcal{X}_i$. Let $\mathbf{x} = (x_1, \cdots, x_p) \in \mathcal{X}$ be a realization of $\mathbf{x}$. In this work, we assume that the random vector $\mathbf{x}$ belongs to an exponential family with bounded support (i.e., length of $\mathcal{X}_i$ is bounded) along with certain additional constraints. More specifically, we make certain assumptions on the natural parameter $\Theta \in \mathbb{R}^{k_1 \times k_2 \times k_3}$, and on the natural statistic $\Phi(\mathbf{x}) : \mathcal{X} \to \mathbb{R}^{k_1 \times k_2 \times k_3}$ as follows.

**Natural parameter $\Theta$.** We focus on natural parameters with bounded norms. However, instead of having such constraints on the natural parameter $\Theta$ as it is, we decompose $\Theta$ into $k_3$ slices (or matrices) and have slice specific constraints. The key motivation for this is to broaden the class of exponential family covered by our formulation. For example, this decomposability allows our formulation to en-capture the sparse-plus-low-rank decomposition of $\Theta$ in addition to only sparse or only low-rank decompositions of $\Theta$ (see Section 2.1). This is precisely the reason for considering tensor natural parameters instead of matrix natural parameters. Further, we assume $k_3 = O(1)$ i.e., it does not scale with $p$. We formally state this assumption below.

**Assumption 2.1.** *(Bounded norms of $\Theta$.) For every $i \in [k_3]$, we let $\mathcal{R}_i(\Theta^{(i)}) \le r_i$ where $\Theta^{(i)} \in \mathbb{R}^{k_1 \times k_2}$ is the $i^{th}$ slice of $\Theta$, $\mathcal{R}_i : \mathbb{R}^{k_1 \times k_2} \to \mathbb{R}_+$ is a norm and $r_i$ is a known constant. This decomposition is represented compactly by $\mathcal{R}(\Theta) \le \mathbf{r}$ where $\mathcal{R}(\Theta) = (\mathcal{R}_1(\Theta^{(1)}), \cdots, \mathcal{R}_{k_3}(\Theta^{(k_3)}))$ and $\mathbf{r} = (r_1, \cdots, r_{k_3})$.*

We define $\Lambda$ to be the set of all natural parameters satisfying Assumption 2.1 i.e., $\Lambda := \{\Theta : \mathcal{R}(\Theta) \le \mathbf{r}\}$. For any $\tilde{\Theta}, \bar{\Theta} \in \Lambda$ and $t \in [0, 1]$, we have $\mathcal{R}(t\tilde{\Theta} + (1-t)\bar{\Theta}) \le t\mathcal{R}(\tilde{\Theta}) + (1-t)\mathcal{R}(\bar{\Theta}) \le t\mathbf{r} + (1-t)\mathbf{r} = \mathbf{r}$. Therefore, $t\tilde{\Theta} + (1-t)\bar{\Theta} \in \Lambda$ and the constraint set $\Lambda$ is a convex set.

---

[3]Even though we focus on continuous variables, our framework applies equally to discrete variables.

**Natural Statistic $\Phi$.** For mathematical simplicity, we center the natural statistic $\Phi(\cdot)$ such that their integral with respect to the uniform density on $\mathcal{X}$ (i.e., $\mathcal{U}_\mathcal{X}$) is zero. $\mathcal{U}_\mathcal{X}$ is well-defined because the support $\mathcal{X}$ is a strict subset of $\mathbb{R}^p$ i.e., $\mathcal{X} \subset \mathbb{R}^p$.

**Definition 2.1.** *(Centered natural statistics). The centered natural statistics are defined as follows:*

$$\Phi(\cdot) := \Phi(\cdot) - \mathbb{E}_{\mathcal{U}_\mathcal{X}}[\Phi(\mathbf{x})].$$

In this work, we focus on bounded natural statistics which may enforce certain restrictions on the length of support $\mathcal{X}$. See Section 2.1 for examples. We define two notions of boundedness. First, we make the following assumption to be able to bound the tensor inner product between the natural parameter $\Theta$ and the centered natural statistic $\Phi(\cdot)$ (see Appendix B.1).

**Assumption 2.2.** *(Bounded dual norms of $\Phi$). For every $i \in [k_3]$ and norm $\mathcal{R}_i$, we assume that the dual norm $\mathcal{R}_i^*$ of the $i^{th}$ slice of the centered natural statistic i.e., $\Phi^{(i)}$ is bounded by a constant $d_i$. Formally, for any $i \in [k_3]$ and $\mathbf{x} \in \mathcal{X}$, $\mathcal{R}_i^*(\Phi^{(i)}(\mathbf{x})) \leq d_i$. This is represented compactly by $\boldsymbol{\mathcal{R}}^*(\Phi(\mathbf{x})) \leq \boldsymbol{d}$ where $\boldsymbol{\mathcal{R}}^*(\Phi(\mathbf{x})) = (\mathcal{R}_1^*(\Phi^{(1)}(\mathbf{x})), \cdots, \mathcal{R}_{k_3}^*(\Phi^{(k_3)}(\mathbf{x})))$ and $\boldsymbol{d} = (d_1, \cdots, d_{k_3})$.*

Next, we assume that the tensor maximum norm of the centered natural statistic $\Phi(\cdot)$ is bounded by a constant $\phi_{\max}$. This assumption is stated formally below.

**Assumption 2.3.** *(Bounded tensor maximum norm of $\Phi$). For any $\mathbf{x} \in \mathcal{X}$, $\|\Phi(\mathbf{x})\|_{\max} \leq \phi_{\max}$.*

**The Exponential Family.** Summarizing, $\mathbf{x}$ belongs to a *minimal truncated* exponential family with probability density function as follows

$$f_\mathbf{x}(\mathbf{x}; \Theta) \propto \exp\left(\left\langle\!\left\langle \Theta, \Phi(\mathbf{x}) \right\rangle\!\right\rangle\right). \tag{9}$$

where the natural parameter $\Theta \in \mathbb{R}^{k_1 \times k_2 \times k_3}$ is such that $\boldsymbol{\mathcal{R}}(\Theta) \leq \boldsymbol{r}$ and the natural statistic $\Phi(\mathbf{x}) : \mathcal{X} \to \mathbb{R}^{k_1 \times k_2 \times k_3}$ is such that for any $\mathbf{x} \in \mathcal{X}$, $\boldsymbol{\mathcal{R}}^*(\Phi(\mathbf{x})) \leq \boldsymbol{d}$ and $\|\Phi(\mathbf{x})\|_{\max} \leq \phi_{\max}$.

Let $\Theta^*$ denote the true natural parameter of interest and $f_\mathbf{x}(\mathbf{x}; \Theta^*)$ denote the true distribution of $\mathbf{x}$. Naturally, we assume $\boldsymbol{\mathcal{R}}(\Theta^*) \leq \boldsymbol{r}$. Formally, the learning task of interest is as follows:

**Goal.** (Natural Parameter Recovery). Given $n$ independent samples of $\mathbf{x}$ i.e., $\mathbf{x}^{(1)} \cdots, \mathbf{x}^{(n)}$ obtained from $f_\mathbf{x}(\mathbf{x}; \Theta^*)$, compute an estimate $\hat{\Theta}$ of $\Theta^*$ in polynomial time such that $\|\Theta^* - \hat{\Theta}\|_T$ is small.

## 2.1 Examples

We will first present examples of natural parameters that satisfy Assumption 2.1. Next, we will present examples of natural statistics along with the corresponding support that satisfy Assumptions 2.2, and 2.3. See Appendix H and I for more discussion on these examples.

**Examples of natural parameter.** We provide examples in Table 1 to illustrate the decomposability of $\Theta$ as in Assumption 2.1. We will revisit these examples briefly in Section 4 and in-depth in Appendix H. Assumption 2.1 should be viewed as a potential flexibility in the problem specification i.e., a practitioner has the option to choose from a variety of constraints on the natural parameters (that could be handled by our framework). For example, in some real-world applications the parameters are sparse while in some other real-world applications the parameters have a low-rank and a practitioner could choose either depending on the application at hand. For the sparse-plus-low-rank decomposition,

Table 1: A few examples of natural parameter $\Theta$.

| Decomposition | $k_3$ | Convex Relaxation |
|---|---|---|
| Sparse decomposition ($\Theta^* = (\Theta^{*(1)})$) | 1 | $\|\Theta^{*(1)}\|_{1,1} \leq r_1$ |
| Low-rank decomposition ($\Theta^* = (\Theta^{*(1)})$) | 1 | $\|\Theta^{*(1)}\|_\star \leq r_1$ |
| Sparse-plus-low-rank decomposition ($\Theta^* = (\Theta^{*(1)}, \Theta^{*(2)})$) | 2 | $\|\Theta^{*(1)}\|_{1,1} \leq r_1$ and $\|\Theta^{*(2)}\|_\star \leq r_2$ |

it is more natural to think about the *minimality* of the exponential family in terms of matrices as opposed to tensors. See Appendix I for details.

**Examples of natural statistic.** The following are a few example of natural statistics (along with the corresponding support) that fall in-line with Assumptions 2.2 and 2.3.

1. *Polynomial statistics*: Suppose the natural statistics are polynomials of $\mathbf{x}$ with maximum degree $l$, i.e., $\prod_{i \in [p]} x_i^{l_i}$ such that $l_i \geq 0 \; \forall i \in [p]$ and $\sum_{i \in [p]} l_i \leq l$. If $\mathcal{X} = [0, b]$ for $b \in \mathbb{R}$, then $\phi_{\max} = 2b^l$. If $\Theta^*$ has a sparse decomposition and $\mathcal{X} = [0, b]$ for $b \in \mathbb{R}$, then $\mathcal{R}^*(\Phi(\mathbf{x})) \leq 2b^k$. Further, if $\Theta^*$ has a low-rank decomposition, $l = 2$, and $\mathcal{X} = \mathcal{B}(0, b)$ for $b \in \mathbb{R}$, then $\mathcal{R}^*(\Phi(\mathbf{x})) \leq 2(1 + b^2)$. Finally, if $\Theta^*$ has a sparse-plus-low-rank decomposition, $l = 2$, and $\mathcal{X} = \mathcal{B}(0, b)$ for $b \in \mathbb{R}$, then $\mathcal{R}^*(\Phi(\mathbf{x})) \leq (2b^2, 2 + 2b^2)$.

2. *Trigonometric statistics*: Suppose the natural statistics are sines and cosines of $\mathbf{x}$ with $l$ different frequencies, i.e., $\sin(\sum_{i \in [p]} l_i x_i) \cup \cos(\sum_{i \in [p]} l_i x_i)$ such that $l_i \in [l] \cup \{0\}$. For any $\mathcal{X} \subset \mathbb{R}^p$, $\phi_{\max} = 2$. If $\Theta^*$ has a sparse decomposition, then $\mathcal{R}^*(\Phi(\mathbf{x})) \leq 2$ for any $\mathcal{X} \subset \mathbb{R}^p$.

Our framework also allows combinations of polynomial and trigonometric statistics (see Appendix I).[4]

## 3 Algorithm

We propose a novel, computationally tractable loss function drawing inspiration from the recent advancements in exponential family Markov Random Fields [56, 55, 44].

**The loss function and the estimator.** The loss function, defined below, is an empirical average of the inverse of the function of $\mathbf{x}$ that the probability density $f_\mathbf{x}(\mathbf{x}; \Theta)$ is proportional to (see (9)).

**Definition 3.1** (The loss function). *Given $n$ samples $\mathbf{x}^{(1)} \cdots, \mathbf{x}^{(n)}$ of $\mathbf{x}$, the loss function maps $\Theta \in \mathbb{R}^{k_1 \times k_2 \times k_3}$ to $\mathcal{L}_n(\Theta) \in \mathbb{R}$ defined as*

$$\mathcal{L}_n(\Theta) = \frac{1}{n} \sum_{t=1}^n \exp\left(-\langle\langle \Theta, \Phi(\mathbf{x}^{(t)}) \rangle\rangle\right). \tag{10}$$

The proposed estimator $\hat{\Theta}_n$ produces an estimate of $\Theta^*$ by minimizing the loss function $\mathcal{L}_n(\Theta)$ over all natural parameters $\Theta$ satisfying Assumption 2.1 i.e.,

$$\hat{\Theta}_n \in \underset{\Theta \in \Lambda}{\arg\min} \, \mathcal{L}_n(\Theta). \tag{11}$$

For any $\epsilon > 0$, $\hat{\Theta}_{\epsilon,n}$ is an $\epsilon$-optimal solution of $\hat{\Theta}_n$ if $\mathcal{L}_n(\hat{\Theta}_{\epsilon,n}) \leq \mathcal{L}_n(\hat{\Theta}_n) + \epsilon$. The optimization in (11) is a convex minimization problem (i.e., minimizing a convex function $\mathcal{L}_n$ over a convex set $\Lambda$) and has efficient implementations for finding an $\epsilon$-optimal solution. Although alternative algorithms (including Frank-Wolfe) can be used, we provide a projected gradient descent algorithm below.

---

**Algorithm 1:** Projected Gradient Descent

**Input:** $\eta, \tau, \Lambda$
**Output:** $\hat{\Theta}_{\epsilon,n}$
**Initialization:** $\Theta_{(0)} = \mathbf{0}$
1 **for** $t = 0, \cdots, \tau$ **do**
2 $\quad \lfloor \; \Theta_{(t+1)} \leftarrow \arg\min_{\Theta \in \Lambda} \|\Theta_{(t)} - \eta \nabla \mathcal{L}_n(\Theta_{(t)}) - \Theta\|_\mathrm{T}$
3 $\hat{\Theta}_{\epsilon,n} \leftarrow \Theta_{(\tau+1)}$

---

The following Lemma shows that running sufficient iterations of the projected gradient descent in Algorithm 1 results in an $\epsilon$-optimal solution of $\hat{\Theta}_n$.

**Lemma 3.1.** *Let Assumptions 2.1, 2.2 and 2.3 be satisfied. Let $\eta = 1/k_1 k_2 k_3 \phi_{\max}^2 \exp(\mathbf{r}^T \mathbf{d})$. Then, Algorithm 1 returns an $\epsilon$-optimal solution $\hat{\Theta}_{\epsilon,n}$ as long as*

$$\tau \geq \frac{2k_1 k_2 k_3 \phi_{\max}^2 \exp(\mathbf{r}^T \mathbf{d})}{\epsilon} \|\hat{\Theta}_n\|_\mathrm{T}^2. \tag{12}$$

*Further, ignoring the dependence on $k_3$, $\phi_{\max}$, $\mathbf{r}$ and $\mathbf{d}$, $\tau$ in (12) scales as $O\left(\mathrm{poly}\left(\frac{k_1 k_2}{\epsilon}\right)\right)$.*

---

[4]We believe that for polynomial and/or trigonometric natural statistics, Assumptions 2.2 and 2.3 would hold whenever the domain of $\mathcal{X}$ is appropriately bounded.

The proof of Lemma 3.1 can be found in Appendix B. The proof outline is as follows : (a) First, we prove the smoothness property of $\mathcal{L}_n(\Theta)$. (b) Next, we complete the proof using a standard result from convex optimization for the projected gradient descent algorithm for smooth functions.

## 4  Analysis and Main results

In this section, we provide our analysis and main results. First, we focus on the connection between our method and the MLE of $f_{\mathbf{x}}(\cdot; \Theta^* - \Theta)$. Then, we establish consistency and asymptotic normality of our estimator. Finally, we provide non-asymptotic finite sample guarantees to recover $\Theta^*$.

**1. Connection with MLE of $f_{\mathbf{x}}(\cdot; \Theta^* - \Theta)$.** First, we will establish a connection between the population version of the loss function in (10) (denoted by $\mathcal{L}(\Theta)$) and the KL-divergence of the uniform density on $\mathcal{X}$ with respect to $f_{\mathbf{x}}(\mathbf{x}; \Theta^* - \Theta)$. Then, using *minimality* of the exponential family, we will show that this KL-divergence and $\mathcal{L}(\Theta)$ are minimized if and only if $\Theta = \Theta^*$. This provides a justification for the estimator in (11) as well as helps us obtain consistency and asymptotic normality of $\hat{\Theta}_n$.

For any $\Theta \in \Lambda$, $\mathcal{L}(\Theta) = \mathbb{E}\Big[\exp\big(-\langle\langle\Theta, \Phi(\mathbf{x})\rangle\rangle\big)\Big]$. The following result shows that the population version of the estimator in (11) is equivalent to the maximum likelihood estimator of $f_{\mathbf{x}}(\mathbf{x}; \Theta^* - \Theta)$.

**Theorem 4.1.** *With $D(\cdot \parallel \cdot)$ representing the KL-divergence,*

$$\arg\min_{\Theta \in \Lambda} \mathcal{L}(\Theta) = \arg\min_{\Theta \in \Lambda} D(\mathcal{U}_{\mathcal{X}}(\cdot) \parallel f_{\mathbf{x}}(\cdot; \Theta^* - \Theta)).$$

*Further, the true parameter $\Theta^*$ is the unique minimizer of $\mathcal{L}(\Theta)$.*

The proof of Theorem 4.1 can be found in Appendix C. The proof outline is as follows : (a) First, we express $f_{\mathbf{x}}(\cdot; \Theta^* - \Theta)$ in terms of $\mathcal{L}(\Theta)$ (b) Next, we complete the proof by simplifying the KL-divergence between $\mathcal{U}_{\mathcal{X}}(\cdot)$ and $f_{\mathbf{x}}(\cdot; \Theta^* - \Theta)$.

**2. Consistency and Normality.** We establish consistency and asymptotic normality of the proposed estimator $\hat{\Theta}_n$ by invoking the asymptotic theory of M-estimation. We emphasize that, from Theorem 4.1, the population version of $\hat{\Theta}_n$ is equivalent to the maximum likelihood estimate of $f_{\mathbf{x}}(\cdot; \Theta^* - \Theta)$ and not $f_{\mathbf{x}}(\cdot; \Theta)$. Moreover, there is no clear connection between $\hat{\Theta}_n$ and the finite sample maximum likelihood estimate of $f_{\mathbf{x}}(\cdot; \Theta)$ or $f_{\mathbf{x}}(\cdot; \Theta^* - \Theta)$. Therefore, we cannot invoke the asymptotic theory of MLE to show consistency and asymptotic normality of $\hat{\Theta}_n$.

Let $A(\Theta^*)$ denote the covariance matrix of $vec\big(\Phi(\mathbf{x}) \exp\big(-\langle\langle\Theta^*, \Phi(\mathbf{x})\rangle\rangle\big)\big)$. Let $B(\Theta^*)$ denote the cross-covariance matrix of $vec(\Phi(\mathbf{x}))$ and $vec(\Phi(\mathbf{x}) \exp\big(-\langle\langle\Theta^*, \Phi(\mathbf{x})\rangle\rangle\big))$. Let $\mathcal{N}(\boldsymbol{\mu}, \boldsymbol{\Sigma})$ represent the multi-variate Gaussian distribution with mean vector $\boldsymbol{\mu}$ and covariance matrix $\boldsymbol{\Sigma}$.

**Theorem 4.2.** *Let Assumptions 2.1, 2.2, and 2.3 be satisfied. Let $\hat{\Theta}_n$ be a solution of (11). Then, as $n \to \infty$, $\hat{\Theta}_n \xrightarrow{p} \Theta^*$. Further, assuming $\Theta^* \in interior(\Lambda)$ and $B(\Theta^*)$ is invertible, we have $\sqrt{n} \times vec(\hat{\Theta}_n - \Theta^*) \xrightarrow{d} \mathcal{N}(vec(\mathbf{0}), B(\Theta^*)^{-1}A(\Theta^*)B(\Theta^*)^{-1})$.*

The proof of Theorem 4.2 can be found in Appendix D. The proof is based on two key observations : (a) $\hat{\Theta}_n$ is an $M$-estimator and (b) $\mathcal{L}(\Theta)$ is uniquely minimized at $\Theta^*$.

**3. Finite Sample Guarantees.** To provide the non-asymptotic guarantees for recovering $\Theta^*$, we require the following assumption on the smallest eigenvalue of the autocorrelation matrix of $vec(\Phi(\mathbf{x}))$.

**Assumption 4.1.** *(Positive eigenvalue of the autocorrelation matrix of $\Phi$.) Let $\lambda_{\min}$ denote the minimum eigenvalue of $\mathbb{E}_{\mathbf{x}}[vec(\Phi(\mathbf{x}))vec(\Phi(\mathbf{x}))^T]$. We assume $\lambda_{\min}$ is strictly positive i.e., $\lambda_{\min} > 0$.*

We also make use of the following property of the matrix norms.

**Property 4.1.** *For any norm $\tilde{\mathcal{R}} : \mathbb{R}^{k_1 \times k_2} \to \mathbb{R}_+$, and matrix $\mathbf{M} \in \mathbb{R}^{k_1 \times k_2}$, there exists $g$ such that $\tilde{\mathcal{R}}(\mathbf{M}) \leq g k_1 k_2 \|\mathbf{M}\|_{\max}$.*

For most matrix norms of interest including entry-wise $L_{p,q}$ norm ($p, q \geq 1$), Schatten $p$-norm ($p \geq 1$), and operator $p-$norm ($p \geq 1$), we have $g = 1$ as shown in Appendix J.

Let $\boldsymbol{g} = (g_1, \cdots, g_{k_3})$ where $\forall i \in [k_3], g_i$ is such that $\mathcal{R}_i^*(\mathbf{M}) \leq g_i k_1 k_2 \|\mathbf{M}\|_{\max}$ with $\mathcal{R}_i^*$ being the dual norms from Assumption 2.2.

Theorem 4.3 below shows that, with enough samples, the $\epsilon$-optimal solution of $\hat{\Theta}_n$ is close to the true natural parameter in the tensor norm with high probability.

**Theorem 4.3.** *Let $\hat{\Theta}_{\epsilon,n}$ be an $\epsilon$-optimal solution of $\hat{\Theta}_n$ obtained from Algorithm 1 for $\epsilon$ of the order $O(\alpha^2 \lambda_{\min})$. Let Assumptions 2.1, 2.2, 2.3, and 4.1 be satisfied. Recall Property 4.1. Then, for any $\delta \in (0, 1)$, we have $\|\hat{\Theta}_{\epsilon,n} - \Theta^*\|_{\mathrm{T}} \leq \alpha$ with probability at least $1 - \delta$ as long as*

$$n \geq O\left( \frac{k_1^2 k_2^2}{\alpha^4 \lambda_{\min}^2} \log\left( \frac{k_1 k_2}{\delta} \right) \right). \tag{13}$$

*The computational cost scales as $O\left( \frac{k_1 k_2}{\alpha^2} \max\left( k_1 k_2 n, c(\Lambda) \right) \right)$ where $c(\Lambda)$ is the cost of projection onto $\Lambda$. Further, ignoring the dependence on $\delta$, $\lambda_{\min}$, and $c(\Lambda)$, $n$ in (13) (as well as the associated computational cost) scales as $O\left( \mathrm{poly}\left( \frac{k_1 k_2}{\alpha} \right) \right)$.*

The proof of Theorem 4.3 can be found in Appendix G. The proof is based on two key properties of the loss function $\mathcal{L}_n(\Theta)$ : (a) with enough samples, the loss function $\mathcal{L}_n(\Theta)$ naturally obeys the restricted strong convexity with high probability and (b) with enough samples, $\|\nabla \mathcal{L}_n(\Theta^*)\|_{\max}$ is bounded with high probability. See the proof for the dependence of the sample complexity and the computational complexity on $k_3, \boldsymbol{r}, \boldsymbol{d}, \boldsymbol{g}$ and $\phi_{\max}$.

The computational cost of projection onto $\Lambda$ i.e., $c(\Lambda)$ is typically polynomial in $k_1 k_2$. In Appendix H, we provide the computational cost for the example constraints on the natural parameter $\Theta$ from Section 2.1 i.e., sparse decomposition, low-rank decomposition, and sparse-plus-low-rank decomposition.

**4. Comparison with the traditional MLE.** To contextualize our method, we compare it with the MLE of the parametric family $f_{\mathbf{x}}(\cdot; \Theta)$. The MLE of $f_{\mathbf{x}}(\cdot; \Theta)$ minimizes the following loss function

$$\min -\frac{1}{n} \sum_{t=1}^{n} \langle\langle \Theta, \Phi(\mathbf{x}^{(t)}) \rangle\rangle + \log \int_{\mathbf{x} \in \mathcal{X}} \exp\left( \langle\langle \Theta, \Phi(\mathbf{x}) \rangle\rangle \right) d\mathbf{x}. \tag{14}$$

The maximum likelihood estimator has many attractive asymptotic properties : (a) consistency (see [16, Theorem 17]), i.e., as the sample size goes to infinity, the bias in the estimated parameters goes to zero, (b) asymptotic normality (see [16, Theorem 18]), i.e., as the sample size goes to infinity, normalized estimation error coverges to a Gaussian distribution and (c) asymptotic efficiency (see [16, Theorem 20]), i.e., as the sample size goes to infinity, the variance in the estimation error attains the minimum possible value among all consistent estimators. Despite having these useful asymptotic properties of consistency, normality, and efficiency, computing the maximum likelihood estimator is computationally hard [52, 26].

Our method can be viewed as a computationally efficient proxy for the MLE. More precisely, our method is computationally tractable as opposed to the MLE while retaining the useful properties of consistency and asymptotic normality. However, our method misses out on asymptotic efficiency. This raises an important question for future work — *can computational and asymptotic efficiency be achieved by a single estimator for this class of exponential family?*

## 5 Conclusion, Remarks, Limitations, Future Work

In this section, we conclude, provide a few remarks, discuss the limitations of our work as well as some interesting future directions.

**Conclusion.** In this work, we provide a computationally and statistically efficient method to learn distributions in a *minimal truncated $k$-parameter exponential family* from i.i.d. samples. We propose a novel estimator via minimizing a convex loss function and obtain consistency and asymptotic normality of the same. We provide rigorous finite sample analysis to achieve an $\alpha$-approximation to the true natural parameters with $O(\mathrm{poly}(k/\alpha))$ samples and $O(\mathrm{poly}(k/\alpha))$ computations. We also provide an interpretation of our estimator in terms of a maximum likelihood estimation.

**Node-wise-sparse exponential family MRFs vs general exponential family.** We highlight that the focus of our work is beyond the exponential families associated with node-wise-sparse MRFs and

towards general exponential families. The former focuses on local assumptions on the parameters such as node-wise-sparsity and the sample complexity depends logarithmically on the parameter dimension i.e., $O(\log(k))$. In contrast, our work can handle global structures on the parameters (e.g., a low-rank constraint) and there are no prior work that can handle such global structures with sample complexity $O(\log(k))$. Similarly, for node-wise-sparse MRFs there has been a lot of work to relax the assumptions required for learning (see the discussion on Assumption 4.1 below). Since our work focuses on global structures associated with the parameters, we leave the question of relaxing the assumptions required for learning as an open question. Likewise, the interaction screening objective [56] and generalized interaction screening objective [55, 44] were designed for node-wise parameter estimation i.e., they require the parameters to be node-wise-sparse and are less useful when the parameters have a global structure. On the contrary, our loss function is designed to accommodate global structures on the parameters.

**Assumption 4.1.** For node-wise-sparse pairwise exponential family MRFs (e.g., Ising models), which is a special case of the setting considered in our work, Assumption 4.1 is proven (e.g., Appendix T.1 of [44] provides one such analysis for a condition that is equivalent to Assumption 4.1 for sparse continuous graphical model). However, such analysis typically requires (a) a bound on the infinity norm of the parameters and a bound on the degree of each node or (b) a bound on the $\ell_1$ norm of the parameters associated with each node. Since the focus of our work is beyond the exponential families associated with node-wise-sparse MRFs, we view Assumption 4.1 as an adequate condition to rule out certain singular distributions (as evident in the proof of Proposition E.1 where this condition is used to effectively lower bounds the variance of a non-constant random variable) and expect it to hold for most real-world applications. Further, we highlight that the MLE in (14) remains computationally intractable even under Assumption 4.1. To see this, one could again focus on node-wise-sparse pairwise exponential family MRFs where Assumption 4.1 is proven and the MLE is still known to be computationally intractable.

**Sample Complexity.** We do not assume $p$ (the dimension of $\mathbf{x}$) to be a constant and think of $k_1$ and $k_2$ as implicit functions of $p$. Typically, for an exponential family, the quantity of interest is the number of parameters i.e., $k$ and this quantity scales polynomially in $p$ e.g., $k = O(p^2)$ for Ising model, $k = O(p^t)$ for t-wise MRFs over binary alphabets. Therefore, in this scenario, the dependence of the sample complexity on $p$ would also be $O(\text{poly}(p))$. Further, the $1/\alpha^4$ dependence of the sample complexity seems fundamental to our loss function. For learning node-wise-sparse MRFs, this dependence is in-line with some prior works that use a similar loss function [44, 55] as well as that do not use a similar loss function [29]. While it is known that for learning node-wise-sparse MRFs [56] and truncated Gaussian [13] one could achieve a better dependence of $1/\alpha^2$, it is not yet clear how the lower bound on the sample complexity would depend on $\alpha$ for the general class of exponential families considered in this work (which may not be sparse or Gaussian).

**Practicality of Algorithm 1.** While the optimization associated with Algorithm 1 is a convex minimization problem (i.e., (11)) and the computational complexity of Algorithm 1 is polynomial in the parameter dimension and the error tolerance, computing the gradient of the loss function requires centering of the natural statistics (see (26)). If the natural statistics are polynomials or trigonometric, centering them should be relatively straightforward (since the integrals would have closed-form expressions). In other cases, centering them may not be polynomial-time and one might require an assumption of computationally efficient sampling or that obtaining approximately random samples of $\mathbf{x}$ is computationally efficient [14].

**Limitations and Future Work.** First, in our current framework, we assume boundedness of the support. While, conceptually, most non-compact distributions could be truncated by introducing a controlled amount of error, we believe this assumption could be lifted as for exponential families: $\mathbb{P}(|x_i| \geq \delta \log \gamma) \leq c\gamma^{-\delta}$ where $c > 0$ is a constant and $\gamma > 0$. Alternatively, the notion of multiplicative regularizing distribution from [41] could also be used. Second, while the population version of our estimator has a nice interpretation in terms of maximum likelihood estimation, the finite sample version of our estimator does not have a similar interpretation. We believe there could be connections with the Bregman score and this is an important direction for immediate future work. Third, while our estimator is computationally efficient, consistent, and asymptotically normal, it is not asymptotically efficient. Investigating the possibility of a single estimator that achieves computational and asymptotic efficiency for this class of exponential family could be an interesting future direction. Lastly, building on our framework, empirical study is an important direction for future work.

## Acknowledgments and Disclosure of Funding

This work was supported, in part, by NSR under Grant No. CCF-1816209, ONR under Grant No. N00014-19-1-2665, the NSF TRIPODS Phase II grant towards Foundations of Data Science Institute, the MIT-IBM project on time series anomaly detection, and the KACST project on Towards Foundations of Reinforcement Learning.

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
