# Appendix

**Organization.** In Appendix A, we provide additional discussion on exponential family Markov random fields, score-based methods, as well as review the related literature on Stein discrepancy and latent variable graphical models. In Appendix B, we state and prove the smoothness property of the loss function as well as provide the proof of Lemma 3.1. In Appendix C, we provide the proof of Theorem 4.1. In Appendix D, we provide the proof of Theorem 4.2. In Appendix E, we provide the restricted strong convexity property of the loss function. In Appendix F, we provide bounds on the tensor maximum norm of the gradient of the loss function evaluated at the true natural parameter. In Appendix G, we provide the proof of Theorem 4.3. In Appendix H, we provide the computational cost for the example constraints on the natural parameter $\Theta$. In Appendix I, we provide a discussion on the examples of natural parameter and natural statistics from Section 2.1. In Appendix J, we provide a discussion on Property 4.1.

**Additional Notations.** We denote the $\ell_p$ norm $(p \geq 1)$ of a vector $\mathbf{v} \in \mathbb{R}^t$ by $\|\mathbf{v}\|_p :=$ $(\sum_{i=1}^t |v_i|^p)^{1/p}$ and its $\ell_\infty$ norm by $\|\mathbf{v}\|_\infty := \max_{i \in [t]} |v_i|$. For a matrix $\mathbf{M} \in \mathbb{R}^{u \times v}$, we denote the spectral norm by $\|\mathbf{M}\| := \max_{i \in [\min\{u,v\}]} \sigma_i(\mathbf{M})$ and the Frobenius norm by $\|\mathbf{M}\|_F :=$ $\sqrt{\sum_{i \in [u], j \in [v]} M_{ij}^2}$. For a tensor $\mathbf{U} \in \mathbb{R}^{u \times v \times w}$, we let $\|\mathbf{U}\|_{1,1,1} := \sum_{i \in [u], j \in [v], l \in [w]} |U_{ijl}|$.

# A   Related Works

In this Section, we review additional works on exponential family Markov random fields, score-based methods, as well as the related literature on Stein discrepancy and latent variable graphical models.

## A.1   Exponential Family Markov Random Fields

Having reviewed some of the works on sparse exponential family MRFs in Section 1.2, we present here a brief overview of a few other works on the same.

Following the lines of [62], the authors in [48] proposed an $\ell_1$ regularized node-conditional log-likelihood to learn the node-conditional density in (5) for non-linear $\phi(\cdot)$. They used an alternating minimization technique and proximal gradient descent to solve the resulting optimization problem. However, their analysis required restricted strong convexity, bounded domain of the variables, non-negative node parameters, and hard-to-verify assumptions on gradient of the population loss.

In [63], the authors introduced a non-parametric component to the node-conditional density in (5) while focusing on linear $\phi(\cdot)$. More specifically, they focused on the following joint density:

$$f_{\mathbf{x}}(\mathbf{x}) \propto \exp\Big( \sum_{i \in [p]} \eta_i(x_i) + \sum_{j \neq i} \theta_{ij} x_i x_j \Big),$$

where $\eta_i(\cdot)$ is the non-parametric node-wise term. They proposed a node-conditional pseudo-likelihood (introduced in [39]) regularized by a non-convex penalty and an adaptive multi-stage convex relaxation method to solve the resulting optimization problem. However, their finite-sample bounds require bounded moments of the variables, sparse eigenvalue condition on their loss function, and local smoothness of the log-partition function. In [49], the authors investigated infinite dimensional sparse pairwise exponential family MRFs where they assumed that the node and edge potentials lie in a Reproducing Kernel Hilbert space (RKHS). They used a penalized version of the score matching objective of [22]. However, their finite-sample analysis required incoherence and dependency conditions (see [59, 24]). In [31], the authors considered the joint distribution in (8) restricting the variables to be non-negative. They proposed a group lasso regularized generalized score matching objective [21] which is a generalization of the score matching objective [22] to non-negative data. However, their finite-sample analysis required the incoherence condition.

## A.2   Score-based and Stein discrepancy methods

Having mentioned the principle behind and an example for the score-based method in Section 1.2, we briefly review a few other score-based methods in relation to the Stein discrepancy.

Stein discrepancy is a quantitative measure of how well a predictive density $q(\cdot)$ fits the density of interest $p(\cdot)$ based on the classical Stein's identity. Stein's identity defines an infinite number of

identities indexed by a critic function $f$ and does not require evaluation of the partition function like the score matching method. By focusing on Stein discrepancy constructed from a RKHS, the authors in [32] and [9] independently proposed the kernel Stein discrepancy as a test statistic to access the goodness-of-fit for unnormalized densities. The authors in [32] and [3] showed that the Fisher divergence, which was the minimization criterion used by the score matching method, can be viewed a special case of the kernel Stein discrepancy with a specific, fixed critic function $f$. In [3], the authors showed that a few other methods (including the contrastive divergence by [19]) can also be viewed as a kernel Stein discrepancy with respect to a different class of critics. Despite the kernel Stein discrepancy being a natural criterion for fitting computationally hard models, there is no clear objective for choosing the right kernel and the kernels typically chosen (e.g. [49, 47, 46, 50] ) are insufficient for complex datasets as pointed out by [61].

In [11], the authors exploited the primal-dual view of the MLE to avoid estimating the normalizing constant at the price of introducing dual variables to be jointly estimated. They showed that many other methods including the contrastive divergence by [19], pseudo-likelihood by [4], score matching by [22] and minimum Stein discrepancy estimator by [32], [9], and [3] are special cases of their estimator. However, this method results in expensive optimization problems since they rely on adversarial optimization (see [42] for details). In [33], the authors proposed an inference method for unnormalized models known as discriminative likelihood estimator. This estimator follows the KL divergence minimization criterion and is implemented via density ratio estimation and a Stein operator. However, this method requires certain hard-to-verify conditions.

### A.3 Literature on Latent Variable Graphical Models

In recent years, sparse-plus-low-rank matrix recovery has received considerable attention in machine learning and statistical inference, e.g., robust PCA [7], latent variable graphical models [8]. Latent variable graphical models has a variety of applications including assessing the functional interactions between neurons recorded from two brain areas [54, 38]. In latent variable graphical models, there are variables not present in observations. The presence of such variables leads to a challenge in learning the graphical model. The graphical model corresponding to the conditional distribution of the observed variables conditioned on the latent variables is in general different from the graphical model corresponding to the marginal distribution of the observed variables. The marginal graphical model consists of dependencies that are induced due to marginalization over the latent variables and typically consists of many more edges than the conditional graphical model. In [8], authors considered latent variable Gaussian graphical models and exploited the observation that the precision matrix of the marginal graphical model can be decomposed into the superposition of a sparse matrix and a low-rank matrix. They provided a tractable convex program based on regularized maximum-likelihood to estimate the precision matrix. While the authors in [8] focused on simultaneous model selection consistency of both the sparse and low-rank components, the authors in [37] focused on estimating the precision matrix of latent variable Gaussian graphical model. They consider a regularized MLE estimator and utilize the *almost strong convexity* [27] of the log-likelihood to derive non-asymptotic error bounds under the restricted Fisher eigenvalue and Structural Fisher Incoherence assumptions. Compared to [37], our tensor norm error bounds are derived under mild condition. Additionally, our framework captures various constraints on the natural parameters in addition to the sparse-plus-low-rank constraint.

## B  Smoothness of the loss function and proof of Lemma 3.1

In this Section, we will prove the smoothness of $\mathcal{L}_n(\cdot)$ as well as prove Lemma 3.1. However, before either of this, we provide bounds on the absolute tensor inner product between $\Theta$ and $\Phi$ i.e., $\left|\langle\langle\Theta, \Phi(\mathbf{x})\rangle\rangle\right|$ for $\Theta \in \Lambda$ and $\mathbf{x} \in \mathcal{X}$.

### B.1  Bounds on the absolute tensor inner product between $\Theta$ and $\Phi$.

We have

$$\left|\langle\langle\Theta, \Phi(\mathbf{x})\rangle\rangle\right| \overset{(a)}{=} \left|\sum_{i=1}^{k_3}\langle\Theta^{(i)}, \Phi^{(i)}(\mathbf{x})\rangle\right| \overset{(b)}{\leq} \sum_{i=1}^{k_3}\left|\langle\Theta^{(i)}, \Phi^{(i)}(\mathbf{x})\rangle\right| \overset{(c)}{\leq} \sum_{i=1}^{k_3}\mathcal{R}_i(\Theta^{(i)}) \times \mathcal{R}_i^*(\Phi^{(i)}(\mathbf{x}))$$

$$\overset{(d)}{\leq} \boldsymbol{r}^T \boldsymbol{d}, \tag{15}$$

where $(a)$ follows from the definitions of a slice of a tensor, tensor inner product, and Frobenius inner product, $(b)$ follows from the triangle inequality, $(c)$ follows from the definition of a dual norm, and $(d)$ follows from Assumptions 2.1 and 2.2.

## B.2 Smoothness of the loss function

Now, we will state and prove our result for smoothness of $\mathcal{L}_n(\Theta)$.

**Proposition B.1.** *Under Assumptions 2.1, 2.2 and 2.3, $\mathcal{L}_n(\Theta)$ is a $k_1 k_2 k_3 \phi_{\max}^2 \exp(\boldsymbol{r}^T \boldsymbol{d})$ smooth function of $\Theta$.*

*Proof of Proposition B.1.* To show $k_1 k_2 k_3 \phi_{\max}^2 \exp(\boldsymbol{r}^T \boldsymbol{d})$ smoothness of $\mathcal{L}_n(\Theta)$, we will show that the largest eigenvalue of the Hessian[5] of $\mathcal{L}_n(\Theta)$ is upper bounded by $k_1 k_2 k_3 \phi_{\max}^2 \exp(\boldsymbol{r}^T \boldsymbol{d})$.

First, we simplify the Hessian of $\mathcal{L}_n(\Theta)$ i.e., $\nabla^2 \mathcal{L}_n(\Theta)$. The component of the Hessian of $\mathcal{L}_n(\Theta)$ corresponding to $\Theta_{u_1 v_1 w_1}$ and $\Theta_{u_2 v_2 w_2}$ for $u_1, u_2 \in [k_1]$, $v_1, v_2 \in [k_2]$ and $w_1, w_2 \in [k_3]$ is given by

$$\frac{\partial^2 \mathcal{L}_n(\Theta)}{\partial \Theta_{u_1 v_1 w_1} \partial \Theta_{u_2 v_2 w_2}} = \frac{1}{n} \sum_{t=1}^{n} \Phi_{u_1 v_1 w_1}(\mathbf{x}^{(t)}) \Phi_{u_2 v_2 w_2}(\mathbf{x}^{(t)}) \exp\left( - \left\langle\!\left\langle \Theta, \Phi(\mathbf{x}^{(t)}) \right\rangle\!\right\rangle \right). \tag{16}$$

From the Gershgorin circle theorem, we know that the largest eigenvalue of any matrix is upper bounded by the largest absolute row sum or column sum. Let $\lambda_{\max}(\nabla^2 \mathcal{L}_n(\Theta))$ denote the largest eigenvalue of $\nabla^2 \mathcal{L}_n(\Theta)$. We have the following

$$\lambda_{\max}(\nabla^2 \mathcal{L}_n(\Theta)) \leq \max_{u_2, v_2, w_2} \sum_{u_1, v_1, w_1} \left| \frac{\partial^2 \mathcal{L}_n(\Theta)}{\partial \Theta_{u_1 v_1 w_1} \partial \Theta_{u_2 v_2 w_2}} \right| \overset{(a)}{\leq} \max_{u_2, v_2, w_2} \sum_{u_1, v_1, w_1} \phi_{\max}^2 \exp(\boldsymbol{r}^T \boldsymbol{d})$$

$$\leq k_1 k_2 k_3 \phi_{\max}^2 \exp(\boldsymbol{r}^T \boldsymbol{d}),$$

where $(a)$ follows from (16), (15), and Assumption 2.3. Therefore, $\mathcal{L}_n(\Theta)$ is a $k_1 k_2 k_3 \phi_{\max}^2 \exp(\boldsymbol{r}^T \boldsymbol{d})$ smooth function of $\Theta$. $\qquad\square$

## B.3 Proof of Lemma 3.1

Next, we restate the Lemma 3.1 and provide the proof.

**Lemma 3.1.** *Let Assumptions 2.1, 2.2 and 2.3 be satisfied. Let $\eta = 1/k_1 k_2 k_3 \phi_{\max}^2 \exp(\boldsymbol{r}^T \boldsymbol{d})$. Then, Algorithm 1 returns an $\epsilon$-optimal solution $\hat{\Theta}_{\epsilon,n}$ as long as*

$$\tau \geq \frac{2 k_1 k_2 k_3 \phi_{\max}^2 \exp(\boldsymbol{r}^T \boldsymbol{d})}{\epsilon} \|\hat{\Theta}_n\|_T^2. \tag{12}$$

*Further, ignoring the dependence on $k_3$, $\phi_{\max}$, $\boldsymbol{r}$ and $\boldsymbol{d}$, $\tau$ in (12) scales as $O\big(\mathrm{poly}\big(\frac{k_1 k_2}{\epsilon}\big)\big)$.*

*Proof of Lemma 3.1.* Let us recall Theorem 10.6 from [35].

[35, Theorem 10.6]: Let $L$ be a $c$-smooth convex function of a parameter vector $\theta \in \Lambda$. Consider the following constrained optimization problem

$$\min_{\theta \in \Lambda} L(\theta). \tag{17}$$

Let $\theta^*$ be an optimal solution of (17). Let $\theta^{(1)}, \cdots, \theta^{(t)}$ denote the iterates of the projected gradient descent algorithm with step size $\eta = 1/c$. Let $\theta^{(0)}$ denote the initialization of $\theta$ in the projected gradient descent algorithm. Then,

$$L(\theta^{(t)}) - L(\theta^*) \leq \frac{2c}{t} \|\theta^{(0)} - \theta^*\|_2^2. \tag{18}$$

---

[5]Ideally, one would consider the Hessian of $\mathcal{L}_n(\mathrm{vec}(\Theta))$. However, for the ease of the exposition we abuse the terminology.

We will make direct use of this theorem in our proof. From Proposition B.1, $\mathcal{L}_n(\Theta)$ is $c_1 := k_1 k_2 k_3 \phi_{\max}^2 \exp(\boldsymbol{r}^T \boldsymbol{d})$ smooth. Using (18), we have

$$\mathcal{L}_n(\Theta_{(\tau)}) - \mathcal{L}_n(\hat{\Theta}_n) \leq \frac{2c_1}{\tau} \|\Theta_{(0)} - \hat{\Theta}_n\|_{\mathrm{T}}^2.$$

Plugging in $c_1 = k_1 k_2 k_3 \phi_{\max}^2 \exp(\boldsymbol{r}^T \boldsymbol{d})$, $\tau = \dfrac{2 k_1 k_2 k_3 \phi_{\max}^2 \exp(\boldsymbol{r}^T \boldsymbol{d})}{\epsilon} \|\hat{\Theta}_n\|_{\mathrm{T}}^2$, and $\Theta_{(0)} = \boldsymbol{0}$ we have

$$\mathcal{L}_n(\Theta_{(\tau)}) - \mathcal{L}_n(\hat{\Theta}_n) \leq \epsilon.$$

Therefore, $\Theta_{(\tau)}$ is an $\epsilon$-optimal solution.

We will now upper bound $\|\hat{\Theta}_n\|_{\mathrm{T}}^2$. First let us upper bound this tensor norm in terms of tensor maximum norm and therefore the matrix maximum norms. We have

$$\|\hat{\Theta}_n\|_{\mathrm{T}}^2 \leq k_1 k_2 k_3 \|\hat{\Theta}_n\|_{\max}^2 = k_1 k_2 k_3 \max_{i \in [k_3]} \|\hat{\Theta}_n^{(i)}\|_{\max}^2.$$

Now, observe that most matrix norms of interest including the entry-wise $L_{p,q}$ norm $(p, q \geq 1)$, the Schatten $p$-norm $(p \geq 1)$, and the operator $p$-norm $(p \geq 1)$ are bounded from below by the matrix maximum norm i.e., the matrix maximum norm is upper bounded if either of these matrix norms are upper bounded. Suppose $\forall i \in [k_3]$, $\mathcal{R}_i$ is either the entry-wise $L_{p,q}$ norm $(p, q \geq 1)$, the Schatten $p$-norm $(p \geq 1)$, or the operator $p$-norm $(p \geq 1)$. Then, $\forall i \in [k_3]$, $\|\hat{\Theta}_n^{(i)}\|_{\max} \leq \mathcal{R}_i(\hat{\Theta}_n^{(i)})$. We have $\mathcal{R}_i(\hat{\Theta}_n^{(i)}) \leq r_i$ from Assumption 2.1 because $\hat{\Theta}_n^{(i)} \in \Lambda$. Therefore, we have

$$\|\hat{\Theta}_n\|_{\mathrm{T}}^2 \leq k_1 k_2 k_3 \max_{i \in [k_3]} r_i^2.$$

Summarizing and using the fact that $\phi_{\max}, \boldsymbol{r}, \boldsymbol{d}, k_3$ are $O(1)$, we have

$$\frac{2 k_1 k_2 k_3 \phi_{\max}^2 \exp(\boldsymbol{r}^T \boldsymbol{d})}{\epsilon} \|\hat{\Theta}_n\|_{\mathrm{T}}^2 \leq \frac{2 k_1^2 k_2^2 k_3^2 \phi_{\max}^2 \exp(\boldsymbol{r}^T \boldsymbol{d})}{\epsilon} \max_{i \in [k_3]} r_i^2 = O\left(\frac{k_1^2 k_2^2}{\epsilon}\right).$$

$\square$

## C   Proof of Theorem 4.1

In this Section, we prove Theorem 4.1. We restate the Theorem below and then provide the proof.

**Theorem 4.1.** *With $D(\cdot \,\|\, \cdot)$ representing the KL-divergence,*

$$\arg\min_{\Theta \in \Lambda} \mathcal{L}(\Theta) = \arg\min_{\Theta \in \Lambda} D(\mathcal{U}_{\mathcal{X}}(\cdot) \,\|\, f_{\mathbf{x}}(\cdot; \Theta^* - \Theta)).$$

*Further, the true parameter $\Theta^*$ is the unique minimizer of $\mathcal{L}(\Theta)$.*

*Proof of Theorem 4.1.* We will first express $f_{\mathbf{x}}(\cdot; \Theta^* - \Theta)$ in terms of $\mathcal{L}(\Theta)$. We have

$$
\begin{aligned}
f_{\mathbf{x}}(\mathbf{x}; \Theta^* - \Theta) &= \frac{\exp\left(\langle\langle \Theta^* - \Theta, \Phi(\mathbf{x}) \rangle\rangle\right)}{\int_{\mathbf{y} \in \mathcal{X}} \exp\left(\langle\langle \Theta^* - \Theta, \Phi(\mathbf{y}) \rangle\rangle\right) d\mathbf{y}} \overset{(a)}{=} \frac{\exp\left(\langle\langle \Theta^* - \Theta, \Phi(\mathbf{x}) \rangle\rangle\right)}{\int_{\mathbf{y} \in \mathcal{X}} \exp\left(\langle\langle \Theta^* - \Theta, \Phi(\mathbf{y}) \rangle\rangle\right) d\mathbf{y}} \\
&\overset{(b)}{=} \frac{f_{\mathbf{x}}(\mathbf{x}; \Theta^*) \exp\left(-\langle\langle \Theta, \Phi(\mathbf{x}) \rangle\rangle\right)}{\int_{\mathbf{y} \in \mathcal{X}} f_{\mathbf{x}}(\mathbf{x}; \Theta^*) \exp\left(-\langle\langle \Theta, \Phi(\mathbf{y}) \rangle\rangle\right) d\mathbf{y}} \\
&\overset{(c)}{=} \frac{f_{\mathbf{x}}(\mathbf{x}; \Theta^*) \exp\left(-\langle\langle \Theta, \Phi(\mathbf{x}) \rangle\rangle\right)}{\mathcal{L}(\Theta)},
\end{aligned}
\tag{19}
$$

where $(a)$ follows because $\mathbb{E}_{\mathcal{U}_{\mathcal{X}}}[\Phi(\mathbf{x})]$ is a constant, $(b)$ follows by dividing the numerator and the denominator by the constant $\int_{\mathbf{y} \in \mathcal{X}} \exp\left(\langle\langle \Theta^*, \Phi(\mathbf{y}) \rangle\rangle\right) d\mathbf{y}$ and using the definition of $f_{\mathbf{x}}(\mathbf{x}; \Theta^*)$, and $(c)$ follows from definition of $\mathcal{L}(\Theta)$. We will now simplify the KL-divergence between $\mathcal{U}_{\mathcal{X}}(\cdot)$ and $f_{\mathbf{x}}(\cdot; \Theta^* - \Theta)$.

$$D(\mathcal{U}_{\mathcal{X}}(\cdot) \,\|\, f_{\mathbf{x}}(\cdot; \Theta^* - \Theta)) \overset{(a)}{=} \mathbb{E}_{\mathcal{U}_{\mathcal{X}}}\left[\log\left(\frac{\mathcal{U}_{\mathcal{X}}(\cdot)\mathcal{L}(\Theta)}{f_{\mathbf{x}}(\cdot; \Theta^*) \exp\left(-\langle\langle \Theta, \Phi(\cdot) \rangle\rangle\right)}\right)\right]$$

$$\stackrel{(b)}{=} \mathbb{E}_{\mathcal{U}_\mathcal{X}} \left[ \log \left( \frac{\mathcal{U}_\mathcal{X}(\cdot)}{f_\mathbf{x}(\cdot; \Theta^*)} \right) \right] + \mathbb{E}_{\mathcal{U}_\mathcal{X}} \left[ \left\langle \left\langle \Theta, \Phi(\cdot) \right\rangle \right\rangle \right] + \log \mathcal{L}(\Theta)$$

$$\stackrel{(c)}{=} \mathbb{E}_{\mathcal{U}_\mathcal{X}} \left[ \log \left( \frac{\mathcal{U}_\mathcal{X}(\cdot)}{f_\mathbf{x}(\cdot; \Theta^*)} \right) \right] + \left\langle \left\langle \Theta, \mathbb{E}_{\mathcal{U}_\mathcal{X}}[\Phi(\cdot)] \right\rangle \right\rangle + \log \mathcal{L}(\Theta)$$

$$\stackrel{(d)}{=} \mathbb{E}_{\mathcal{U}_\mathcal{X}} \left[ \log \left( \frac{\mathcal{U}_\mathcal{X}(\cdot)}{f_\mathbf{x}(\cdot; \Theta^*)} \right) \right] + \log \mathcal{L}(\Theta),$$

where $(a)$ follows from (19) and the definition of KL-divergence, $(b)$ follows because $\log(abc) = \log a + \log b + \log c$ and $\mathcal{L}(\Theta)$ is a constant, $(c)$ follows from the linearity of the expectation and $(d)$ follows because $\mathbb{E}_{\mathcal{U}_\mathcal{X}}[\Phi(\mathbf{x})] = 0$ from Definition 2.1. Observing that the first term in the above equation is not dependent on $\Theta$, we can write

$$\arg\min_{\Theta \in \Lambda} D(\mathcal{U}_\mathcal{X}(\cdot) \parallel f_\mathbf{x}(\cdot; \Theta^* - \Theta)) = \arg\min_{\Theta \in \Lambda} \log \mathcal{L}(\Theta) \stackrel{(a)}{=} \arg\min_{\Theta \in \Lambda} \mathcal{L}(\Theta),$$

where $(a)$ follows because $\log$ is a monotonic function. Further, the KL-divergence between $\mathcal{U}_\mathcal{X}(\cdot)$ and $f_\mathbf{x}(\cdot; \Theta^* - \Theta)$ is minimized when $\mathcal{U}_\mathcal{X}(\cdot) = f_\mathbf{x}(\cdot; \Theta^* - \Theta)$. Recall that the natural statistic are such that the exponential family is minimal. Therefore, $\mathcal{U}_\mathcal{X}(\cdot) = f_\mathbf{x}(\cdot; \Theta^* - \Theta)$ if and only if $\Theta = \Theta^*$. Thus, $\Theta^* \in \arg\min_{\Theta \in \Lambda} \mathcal{L}(\Theta)$, and it is a unique minimizer of $\mathcal{L}(\Theta)$. $\qquad\square$

## D   Proof of Theorem 4.2

In this Section, we prove Theorem 4.2 by using the theory of $M$-estimation. In particular, observe that $\hat{\Theta}_n$ is an $M$-estimator i.e., $\hat{\Theta}_n$ is a sample average. Therefore, we invoke Theorem 4.1.1 and Theorem 4.1.3 of [1] to prove the consistency and normality of $\hat{\Theta}_n$. We restate the Theorem below and then provide the proof.

**Theorem 4.2.** *Let Assumptions 2.1, 2.2, and 2.3 be satisfied. Let $\hat{\Theta}_n$ be a solution of* (11). *Then, as $n \to \infty$, $\hat{\Theta}_n \stackrel{p}{\to} \Theta^*$. Further, assuming $\Theta^* \in interior(\Lambda)$ and $B(\Theta^*)$ is invertible, we have $\sqrt{n} \times vec(\hat{\Theta}_n - \Theta^*) \stackrel{d}{\to} \mathcal{N}(vec(\mathbf{0}), B(\Theta^*)^{-1} A(\Theta^*) B(\Theta^*)^{-1})$.*

*Proof of Theorem 4.2.* We divide the proof in two parts.

**Consistency.** We will first show that $\hat{\Theta}_n$ is asymptotically consistent. In order to show this, let us recall Theorem 4.1.1 of [1].

[1, Theorem 4.1.1]: Let $z_1, \cdots, z_n$ be i.i.d. samples of a random variable $z$. Let $q(z; \theta)$ be some function of $z$ parameterized by $\theta \in \Upsilon$. Let $\theta^*$ be the true underlying parameter. Define

$$Q_n(\theta) = \frac{1}{n} \sum_{i=1}^{n} q(z_i; \theta) \qquad \text{and} \qquad \hat{\theta}_n \in \arg\min_{\theta \in \Upsilon} Q_n(\theta).$$

Let the following be true.

(a) $\Upsilon$ is compact,

(b) $Q_n(\theta)$ converges uniformly in probability to a non-stochastic function $Q(\theta)$,

(c) $Q(\theta)$ is continuous, and

(d) $Q(\theta)$ is uniquely minimized at $\theta^*$.

Then, $\hat{\theta}_n$ is consistent for $\theta^*$ i.e., $\hat{\theta}_n \stackrel{p}{\to} \theta^*$ as $n \to \infty$.

Letting $z := \mathbf{x}$, $\theta := \Theta$, $\hat{\theta}_n := \hat{\Theta}_n$, $\theta^* := \Theta^*$, $\Upsilon = \Lambda$, $q(z; \theta) := \exp\left( - \left\langle \left\langle \Theta, \Phi(\mathbf{x}) \right\rangle \right\rangle \right)$, and $Q_n(\theta) := \mathcal{L}_n(\Theta)$, it is sufficient to show the following:

(a) $\Lambda$ is compact,

(b) $\mathcal{L}_n(\Theta)$ converges uniformly in probability to a non-stochastic function $\mathcal{L}(\Theta)$,

(c) $\mathcal{L}(\Theta)$ is continuous, and

(d) $\mathcal{L}(\Theta)$ is uniquely minimized at $\Theta^*$.

Let us show these one by one.

(a) We have $\Lambda = \{\Theta : \mathcal{R}(\Theta) \leq r\}$ which is bounded and closed. Therefore, $\Lambda$ is compact.

(b) Recall [25, Theorem 2]: Let $z_1, \cdots, z_n$ be i.i.d. samples of a random variable $z$. Let $g(z; \theta)$ be a function of $\theta$ parameterized by $\theta \in \Upsilon$. Then, $n^{-1} \sum_t g(z_t, \theta)$ converges uniformly in probability to $\mathbb{E}[g(z, \theta)]$ if

   (i) $\Upsilon$ is compact,

   (ii) $g(z, \theta)$ is continuous at each $\theta \in \Upsilon$ with probability one,

   (iii) $g(z, \theta)$ is dominated by a function $G(z)$ i.e., $|g(z, \theta)| \leq G(z)$, and

   (iv) $\mathbb{E}[G(z)] < \infty$.

Using this theorem with $z := \mathbf{x}$, $\theta := \Theta$, $\Upsilon := \Lambda$, $g(z, \theta) := \exp\left(-\langle\langle \Theta, \Phi(\mathbf{x}) \rangle\rangle\right)$, $G(z) := \exp(\mathbf{r}^T \mathbf{d})$ and (15), we conclude that $\mathcal{L}_n(\Theta)$ converges to $\mathcal{L}(\Theta)$ uniformly in probability.

(c) $\exp\left(-\langle\langle \Theta, \Phi(\mathbf{x}) \rangle\rangle\right)$ is a continuous function of $\Theta \in \Lambda$. Further, $f_{\mathbf{x}}(\mathbf{x}; \Theta^*)$ does not functionally depend on $\Theta$. Therefore, we have continuity of $\mathcal{L}(\Theta)$ for all $\Theta \in \Lambda$.

(d) From Theorem 4.1, $\mathcal{L}(\Theta)$ is uniquely minimized at $\Theta^*$.

Therefore, we have asymptotic consistency of $\hat{\Theta}_n$.

**Normality.** We will now show that $\hat{\Theta}_n$ is asymptotically normal. In order to show this, let us recall Theorem 4.1.3 of [1].

[1, Theorem 4.1.3]: Let $z_1, \cdots, z_n$ be i.i.d. samples of a random variable $z$. Let $q(z; \theta)$ be some function of $z$ parameterized by $\theta \in \Upsilon$. Let $\theta^*$ be the true underlying parameter. Define

$$Q_n(\theta) = \frac{1}{n} \sum_{i=1}^{n} q(z_i; \theta) \qquad \text{and} \qquad \hat{\theta}_n \in \underset{\theta \in \Upsilon}{\arg\min}\, Q_n(\theta).$$

Let the following be true.

(a) $\hat{\theta}_n$ is consistent for $\theta^*$,

(b) $\theta^*$ lies in the interior of the parameter space $\Upsilon$,

(c) $Q_n$ is twice continuously differentiable in an open and convex neighborhood of $\theta^*$,

(d) $\sqrt{n} \nabla Q_n(\theta)|_{\theta = \theta^*} \xrightarrow{d} \mathcal{N}(\mathbf{0}, A(\theta^*))$, and

(e) $\nabla^2 Q_n(\theta)|_{\theta = \hat{\theta}_n} \xrightarrow{p} B(\theta^*)$ with $B(\theta)$ finite, non-singular, and continuous at $\theta^*$,

Then, $\hat{\theta}_n$ is normal for $\theta^*$ i.e., $\sqrt{n}(\hat{\theta}_n - \theta^*) \xrightarrow{d} \mathcal{N}(\mathbf{0}, B^{-1}(\theta^*)A(\theta^*)B^{-1}(\theta^*))$.

Letting $z := \mathbf{x}$, $\theta := \Theta$, $\hat{\theta}_n := \hat{\Theta}_n$, $\theta^* := \Theta^*$, $\Upsilon = \Lambda$, $q(z; \theta) := \exp\left(-\langle\langle \Theta, \Phi(\mathbf{x}) \rangle\rangle\right)$, and $Q_n(\theta) := \mathcal{L}_n(\Theta)$, it is sufficient to show the following:

(a) $\hat{\Theta}_n$ is consistent for $\Theta^*$,

(b) $\Theta^*$ lies in the interior of the parameter space $\Lambda$,

(c) $\mathcal{L}_n$ is twice continuously differentiable in an open and convex neighborhood of $\Theta^*$,

(d) $\sqrt{n} \nabla \mathcal{L}_n(\text{vec}(\Theta))|_{\Theta = \Theta^*} \xrightarrow{d} \mathcal{N}(\mathbf{0}, A(\Theta^*))$, and

(e) $\nabla^2 \mathcal{L}_n(\text{vec}(\Theta))|_{\Theta = \hat{\Theta}_n} \xrightarrow{p} B(\Theta^*)$ with $B(\Theta)$ finite, non-singular, and continuous at $\Theta^*$,

Let us show these one by one.

(a) We have established that $\hat{\Theta}_n$ is consistent for $\Theta^*$ in the first half of the proof.

(b) The assumption that $\Theta^* \in \text{interior}(\Lambda)$ is equivalent to $\Theta^*$ belonging to the interior of $\Lambda$.

(c) Fix $u_1, u_2 \in [k_1]$, $v_1, v_2 \in [k_2]$, and $w_1, w_2 \in [k_3]$. We have

$$\frac{\partial^2 \mathcal{L}_n(\Theta)}{\partial \Theta_{u_1 v_1 w_1} \partial \Theta_{u_2 v_2 w_2}} = \frac{1}{n} \sum_{t=1}^n \Phi_{u_1 v_1 w_1}(\mathbf{x}^{(t)}) \Phi_{u_2 v_2 w_2}(\mathbf{x}^{(t)}) \exp\left(-\langle\langle\Theta, \Phi(\mathbf{x}^{(t)})\rangle\rangle\right).$$

Thus, $\partial^2 \mathcal{L}_n(\Theta)/\partial\Theta_{u_1 v_1 w_1}\partial\Theta_{u_2 v_2 w_2}$ exists. Using the continuity of $\Phi(\cdot)$ and $\exp\left(-\langle\langle\Theta, \Phi(\cdot)\rangle\rangle\right)$, we see that $\partial^2 \mathcal{L}_n(\Theta)/\partial\Theta_{u_1 v_1 w_1}\partial\Theta_{u_2 v_2 w_2}$ is continuous in an open and convex neighborhood of $\Theta^*$.

(d) For any $u \in [k_1]$, $v \in [k_2]$ and $w \in [k_3]$, define the random variable

$$\chi_{uvw} = -\Phi_{uvw}(\mathbf{x}) \exp\left(-\langle\langle\Theta^*, \Phi(\mathbf{x})\rangle\rangle\right).$$

The component of the gradient of $\mathcal{L}_n(\text{vec}(\Theta))$ corresponding to $\Theta_{uvw}$ evaluated at $\Theta^*$ is given by

$$\frac{\partial \mathcal{L}_n(\Theta^*)}{\partial \Theta_{uvw}} = -\frac{1}{n} \sum_{t=1}^n \Phi_{uvw}(\mathbf{x}^{(t)}) \exp\left(-\langle\langle\Theta^*, \Phi(\mathbf{x}^{(t)})\rangle\rangle\right).$$

Each term in the above summation is distributed as the random variable $\chi_{uvw}$. The random variable $\chi_{uvw}$ has zero mean (see Lemma F.1). Using this and the multivariate central limit theorem [53], we have

$$\sqrt{n}\nabla\mathcal{L}_n(\text{vec}(\Theta))|_{\Theta=\Theta^*} \xrightarrow{d} \mathcal{N}(\mathbf{0}, A(\Theta^*)),$$

where $A(\Theta^*)$ is the covariance matrix of $\text{vec}\left(\Phi(\mathbf{x}) \exp\left(-\langle\langle\Theta^*, \Phi(\mathbf{x})\rangle\rangle\right)\right)$.

(e) We will start by showing that the following is true.

$$\nabla^2 \mathcal{L}_n(\text{vec}(\Theta))|_{\Theta=\hat{\Theta}_n} \xrightarrow{p} \nabla^2 \mathcal{L}(\text{vec}(\Theta))|_{\Theta=\Theta^*}. \tag{20}$$

To begin with, using the uniform law of large numbers [25, Theorem 2] for any $\Theta \in \Lambda$ results in

$$\nabla^2 \mathcal{L}_n(\text{vec}(\Theta)) \xrightarrow{p} \nabla^2 \mathcal{L}(\text{vec}(\Theta)). \tag{21}$$

Using the consistency of $\hat{\Theta}_n$ and the continuous mapping theorem, we have

$$\nabla^2 \mathcal{L}(\text{vec}(\Theta))|_{\Theta=\hat{\Theta}_n} \xrightarrow{p} \nabla^2 \mathcal{L}(\text{vec}(\Theta))|_{\Theta=\Theta^*}. \tag{22}$$

Let $u_1, u_2 \in [k_1]$, $v_1, v_2 \in [k_2]$, and $w_1, w_2 \in [k_3]$. From (21) and (22), for any $\epsilon > 0$, for any $\delta > 0$, there exists integers $n_1, n_2$ such that for $n \geq \max\{n_1, n_2\}$ we have,

$$\mathbb{P}(|\partial^2 \mathcal{L}_n(\hat{\Theta}_n)/\partial\Theta_{u_1 v_1 w_1}\partial\Theta_{u_2 v_2 w_2} - \partial^2 \mathcal{L}(\hat{\Theta}_n)/\partial\Theta_{u_1 v_1 w_1}\partial\Theta_{u_2 v_2 w_2}| > \epsilon/2) \leq \delta/2$$

and

$$\mathbb{P}(|\partial^2 \mathcal{L}(\hat{\Theta}_n)/\partial\Theta_{u_1 v_1 w_1}\partial\Theta_{u_2 v_2 w_2} - \partial^2 \mathcal{L}(\Theta^*)/\partial\Theta_{u_1 v_1 w_1}\partial\Theta_{u_2 v_2 w_2}| > \epsilon/2) \leq \delta/2.$$

Now for $n \geq \max\{n_1, n_2\}$, using the triangle inequality we have

$$\mathbb{P}(|\partial^2 \mathcal{L}_n(\hat{\Theta}_n)/\partial\Theta_{u_1 v_1 w_1}\partial\Theta_{u_2 v_2 w_2} - \partial^2 \mathcal{L}(\Theta^*)/\partial\Theta_{u_1 v_1 w_1}\partial\Theta_{u_2 v_2 w_2}| > \epsilon) \leq \delta/2 + \delta/2 = \delta.$$

Thus, we have (20). Using the definition of $\mathcal{L}(\Theta)$, we have

$$\partial^2 \mathcal{L}(\Theta^*)/\partial\Theta_{u_1 v_1 w_1}\partial\Theta_{u_2 v_2 w_2} = \mathbb{E}\left[\Phi_{u_1 v_1 w_1}(\mathbf{x})\Phi_{u_2 v_2 w_2}(\mathbf{x}) \exp\left(-\langle\langle\Theta^*, \Phi(\mathbf{x})\rangle\rangle\right)\right]$$

$$\stackrel{(b)}{=} \mathbb{E}\left[\Phi_{u_1 v_1 w_1}(\mathbf{x})\Phi_{u_2 v_2 w_2}(\mathbf{x}) \exp\left(-\langle\langle\Theta^*, \Phi(\mathbf{x})\rangle\rangle\right)\right]$$

$$- \mathbb{E}\Big[\Phi_{u_1 v_1 w_1}(\mathbf{x})\Big]\mathbb{E}\Big[\Phi_{u_2 v_2 w_2}(\mathbf{x}) \exp\big(-\langle\langle\Theta^*, \Phi(\mathbf{x})\rangle\rangle\big)\Big]$$

$$= \mathrm{cov}\Big(\Phi_{u_1 v_1 w_1}(\mathbf{x}), \Phi_{u_2 v_2 w_2}(\mathbf{x}) \exp\big(-\langle\langle\Theta^*, \Phi(\mathbf{x})\rangle\rangle\big)\Big),$$

where (b) follows because $\mathbb{E}\big[\Phi_{u_2 v_2 w_2}(\mathbf{x}) \exp\big(-\langle\langle\Theta^*, \Phi(\mathbf{x})\rangle\rangle\big)\big] = 0$ for any $u_2 \in [k_1]$, $v_2 \in [k_2]$, and $w_2 \in [k_3]$ from Lemma F.1. Therefore, we have

$$\nabla^2 \mathcal{L}_n(\mathrm{vec}(\Theta))|_{\Theta=\hat{\Theta}_n} \xrightarrow{p} B(\Theta^*),$$

where $B(\Theta^*)$ is the cross-covariance matrix of $\mathrm{vec}(\Phi(\mathbf{x}))$ and $\mathrm{vec}\big(\Phi(\mathbf{x}) \exp\big(-\langle\langle\Theta^*, \Phi(\mathbf{x})\rangle\rangle\big)\big)$. Finiteness and continuity of $\Phi(\mathbf{x})$ and $\Phi(\mathbf{x}) \exp\big(-\langle\langle\Theta^*, \Phi(\mathbf{x})\rangle\rangle\big)$ implies the finiteness and continuity of $B(\Theta^*)$. By assumption, the cross-covariance matrix of $\mathrm{vec}(\Phi(\mathbf{x}))$ and $\mathrm{vec}\big(\Phi(\mathbf{x}) \exp\big(-\langle\langle\Theta^*, \Phi(\mathbf{x})\rangle\rangle\big)\big)$ is invertible.

Therefore, we have the asymptotic normality of $\hat{\Theta}_n$. $\qquad\qquad\square$

# E   Restricted strong convexity of the loss function

In this Section, we will show that, with enough samples, the loss function obeys the restricted strong convexity property with high probability. This result will in turn allow us to prove Theorem 4.3 in Appendix G

We will first state the main result of this Section (Proposition E.1). Next, we will introduce the notion of correlation for the centered natural statistics and provide a supporting Lemma wherein we will bound the deviation between the true correlation and the empirical correlation. Finally, we will prove Proposition E.1.

Consider any $\Theta \in \Lambda$. Let $\Delta = \Theta - \Theta^*$. Define the residual of the first-order Taylor expansion as

$$\delta\mathcal{L}_n(\Delta, \Theta^*) = \mathcal{L}_n(\Theta^* + \Delta) - \mathcal{L}_n(\Theta^*) - \langle\langle\nabla\mathcal{L}_n(\Theta^*), \Delta\rangle\rangle. \tag{23}$$

**Proposition E.1.** *Let Assumptions 2.1, 2.2, 2.3 and 4.1 be satisfied. For any $\delta_3 \in (0, 1)$, the residual defined in* (23) *satisfies*

$$\delta\mathcal{L}_n(\Delta, \Theta^*) \geq \frac{\lambda_{\min} \exp(-\boldsymbol{r}^T \boldsymbol{d})}{4(1 + \boldsymbol{r}^T \boldsymbol{d})}\|\Delta\|_{\mathrm{T}}^2,$$

*with probability at least $1 - \delta_3$ as long as*

$$n > \frac{8\phi_{\max}^4 k_1^2 k_2^2 k_3^3}{\lambda_{\min}^2} \log\Big(\frac{2k_1^2 k_2^2 k_3^3}{\delta_3}\Big).$$

## E.1   Correlation between centered natural statistics

For any $u_1, u_2 \in [k_1]$, $v_1, v_2 \in [k_2]$, and $w_1, w_2 \in [k_3]$, let $H_{u_1 v_1 w_1 u_2 v_2 w_2}$ denote the correlation between $\Phi_{u_1 v_1 w_1}(\mathbf{x})$ and $\Phi_{u_2 v_2 w_2}(\mathbf{x})$ defined as

$$H_{u_1 v_1 w_1 u_2 v_2 w_2} = \mathbb{E}\big[\Phi_{u_1 v_1 w_1}(\mathbf{x})\Phi_{u_2 v_2 w_2}(\mathbf{x})\big], \tag{24}$$

and let $\mathbf{H} = [H_{u_1 v_1 w_1 u_2 v_2 w_2}] \in \mathbb{R}^{[k_1] \times [k_2] \times [k_3] \times [k_1] \times [k_2] \times [k_3]}$ be the corresponding correlation tensor. Similarly, we define $\hat{\mathbf{H}}$ based on the empirical estimates of the correlation

$$\hat{H}_{u_1 v_1 w_1 u_2 v_2 w_2} = \frac{1}{n}\sum_{t=1}^{n} \Phi_{u_1 v_1 w_1}(\mathbf{x}^{(t)})\Phi_{u_2 v_2 w_2}(\mathbf{x}^{(t)}). \tag{25}$$

The following lemma bounds the deviation between the true correlation and the empirical correlation.

**Lemma E.1.** *Consider any $u_1, u_2 \in [k_1]$, $v_1, v_2 \in [k_2]$, and $w_1, w_2 \in [k_3]$. Let Assumption 2.3 be satisfied. Then, we have for any $\epsilon_2 > 0$,*

$$|\hat{H}_{u_1 v_1 w_1 u_2 v_2 w_2} - H_{u_1 v_1 w_1 u_2 v_2 w_2}| < \epsilon_2,$$

*with probability at least $1 - \delta_2$ as long as*

$$n > \frac{2\phi_{\max}^4}{\epsilon_2^2} \log\left(\frac{2k_1^2 k_2^2 k_3^2}{\delta_2}\right).$$

*Proof of Lemma E.1.* Fix $u_1, u_2 \in [k_1]$, $v_1, v_2 \in [k_2]$, and $w_1, w_2 \in [k_3]$. The random variable defined as $Y_{u_1 v_1 w_1 u_2 v_2 w_2} := \Phi_{u_1 v_1 w_1}(\mathbf{x}) \Phi_{u_2 v_2 w_2}(\mathbf{x})$ satisfies $|Y_{u_1 v_1 w_1 u_2 v_2 w_2}| \leq \phi_{\max}^2$ (from Assumption 2.3). Using the Hoeffding inequality we get

$$\mathbb{P}\left(|\hat{H}_{u_1 v_1 w_1 u_2 v_2 w_2} - H_{u_1 v_1 w_1 u_2 v_2 w_2}| > \epsilon_2\right) < 2\exp\left(-\frac{n\epsilon_2^2}{2\phi_{\max}^4}\right).$$

The proof follows by using the union bound over all $u_1, u_2 \in [k_1]$, $v_1, v_2 \in [k_2]$, and $w_1, w_2 \in [k_3]$. $\qquad\square$

## E.2 Proof of Proposition E.1

*Proof of Proposition E.1.* First, we will simplify the gradient of $\mathcal{L}_n(\Theta)$[6] evaluated at $\Theta^*$. For any $u \in [k_1]$, $v \in [k_2]$ and $w \in [k_3]$, the component of the gradient of $\mathcal{L}_n(\Theta)$ corresponding to $\Theta_{uvw}$ evaluated at $\Theta^*$ is given by

$$\frac{\partial \mathcal{L}_n(\Theta^*)}{\partial \Theta_{uvw}} = -\frac{1}{n}\sum_{t=1}^n \Phi_{uvw}(\mathbf{x}^{(t)}) \exp\left(-\langle\langle\Theta^*, \Phi(\mathbf{x}^{(t)})\rangle\rangle\right). \tag{26}$$

We will now provide the desired lower bound on the residual. Substituting (10) and (26) in (23), we have

$$\delta\mathcal{L}_n(\Delta, \Theta^*) = \frac{1}{n}\sum_{t=1}^n \exp\left(-\langle\langle\Theta^*, \Phi(\mathbf{x}^{(t)})\rangle\rangle\right) \times \left[\exp\left(-\langle\langle\Delta, \Phi(\mathbf{x}^{(t)})\rangle\rangle\right) - 1 + \langle\langle\Delta, \Phi(\mathbf{x}^{(t)})\rangle\rangle\right]$$

$$\overset{(a)}{\geq} \exp(-\mathbf{r}^T\mathbf{d}) \times \frac{1}{n}\sum_{t=1}^n \left[\exp\left(-\langle\langle\Delta, \Phi(\mathbf{x}^{(t)})\rangle\rangle\right) - 1 + \langle\langle\Delta, \Phi(\mathbf{x}^{(t)})\rangle\rangle\right]$$

$$\overset{(b)}{\geq} \exp(-\mathbf{r}^T\mathbf{d}) \times \frac{1}{n}\sum_{t=1}^n \frac{|\langle\langle\Delta, \Phi(\mathbf{x}^{(t)})\rangle\rangle|^2}{2 + |\langle\langle\Delta, \Phi(\mathbf{x}^{(t)})\rangle\rangle|}$$

$$\overset{(c)}{\geq} \frac{\exp(-\mathbf{r}^T\mathbf{d})}{2 + 2\mathbf{r}^T\mathbf{d}} \times \frac{1}{n}\sum_{t=1}^n |\langle\langle\Delta, \Phi(\mathbf{x}^{(t)})\rangle\rangle|^2$$

$$\overset{(d)}{=} \frac{\exp(-\mathbf{r}^T\mathbf{d})}{2 + 2\mathbf{r}^T\mathbf{d}} \times \sum_{u_1=1}^{k_1}\sum_{v_1=1}^{k_2}\sum_{w_1=1}^{k_3}\sum_{u_2=1}^{k_1}\sum_{v_2=1}^{k_2}\sum_{w_2=1}^{k_3} \Delta_{u_1 v_1 w_1} \hat{H}_{u_1 v_1 w_1 u_2 v_2 w_2} \Delta_{u_2 v_2 w_2}$$

$$= \frac{\exp(-\mathbf{r}^T\mathbf{d})}{2 + 2\mathbf{r}^T\mathbf{d}} \times \sum_{u_1=1}^{k_1}\sum_{v_1=1}^{k_2}\sum_{w_1=1}^{k_3}\sum_{u_2=1}^{k_1}\sum_{v_2=1}^{k_2}\sum_{w_2=1}^{k_3} \Delta_{u_1 v_1 w_1} \times$$

$$[H_{u_1 v_1 w_1 u_2 v_2 w_2} + \hat{H}_{u_1 v_1 w_1 u_2 v_2 w_2} - H_{u_1 v_1 w_1 u_2 v_2 w_2}]\Delta_{u_2 v_2 w_2},$$

where $(a)$ follows because $-\langle\langle\Theta, \Phi(\mathbf{x})\rangle\rangle \geq -\mathbf{r}^T\mathbf{d}$ from (15), $(b)$ follows because $e^{-z} - 1 + z \geq \frac{z^2}{2 + |z|}$ for any $z \in \mathbb{R}$, $(c)$ follows from (15), and $(d)$ follows from (25).

---

[6]Ideally, one would consider the gradient of $\mathcal{L}_n(\text{vec}(\Theta))$. However, for the ease of the exposition we abuse the terminology.

Let the number of samples satisfy

$$n > \frac{8\phi_{\max}^4 k_1^2 k_2^2 k_3^2}{\lambda_{\min}^2} \log\left(\frac{2k_1^2 k_2^2 k_3^2}{\delta_3}\right).$$

Using Lemma E.1 with $\epsilon_2 = \frac{\lambda_{\min}}{2k_1 k_2 k_3}$ and $\delta_2 = \delta_3$, and the triangle inequality, we have the following with probability at least $1 - \delta_3$

$$\delta\mathcal{L}_n(\Delta, \Theta^*) \geq \frac{\exp(-\boldsymbol{r}^T \boldsymbol{d})}{2 + 2\boldsymbol{r}^T \boldsymbol{d}} \times \Bigg[ \sum_{u_1=1}^{k_1} \sum_{v_1=1}^{k_2} \sum_{w_1=1}^{k_3} \sum_{u_2=1}^{k_1} \sum_{v_2=1}^{k_2} \sum_{w_2=1}^{k_3} \Delta_{u_1 v_1 w_1} H_{u_1 v_1 w_1 u_2 v_2 w_2} \Delta_{u_2 v_2 w_2}$$

$$- \frac{\lambda_{\min}}{2k_1 k_2 k_3} \|\Delta\|_{1,1,1}^2 \Bigg]$$

$$\overset{(a)}{\geq} \frac{\exp(-\boldsymbol{r}^T \boldsymbol{d})}{2 + 2\boldsymbol{r}^T \boldsymbol{d}} \times \Bigg[ \sum_{u_1=1}^{k_1} \sum_{v_1=1}^{k_2} \sum_{w_1=1}^{k_3} \sum_{u_2=1}^{k_1} \sum_{v_2=1}^{k_2} \sum_{w_2=1}^{k_3} \Delta_{u_1 v_1 w_1} H_{u_1 v_1 w_1 u_2 v_2 w_2} \Delta_{u_2 v_2 w_2}$$

$$- \frac{\lambda_{\min}}{2} \|\Delta\|_{\mathrm{T}}^2 \Bigg]$$

$$\overset{(b)}{=} \frac{\exp(-\boldsymbol{r}^T \boldsymbol{d})}{2 + 2\boldsymbol{r}^T \boldsymbol{d}} \times \Bigg[ \mathrm{vec}(\Delta)\mathbb{E}[\mathrm{vec}(\varPhi(\mathbf{x}))\mathrm{vec}(\varPhi(\mathbf{x}))^T]\mathrm{vec}(\Delta)^T - \frac{\lambda_{\min}}{2} \|\Delta\|_{\mathrm{T}}^2 \Bigg]$$

$$\overset{(c)}{\geq} \frac{\exp(-\boldsymbol{r}^T \boldsymbol{d})}{2 + 2\boldsymbol{r}^T \boldsymbol{d}} \times \Bigg[ \lambda_{\min} \|\mathrm{vec}(\Delta)\|_2^2 - \frac{\lambda_{\min}}{2} \|\Delta\|_{\mathrm{T}}^2 \Bigg]$$

$$\overset{(d)}{=} \frac{\exp(-\boldsymbol{r}^T \boldsymbol{d})}{2 + 2\boldsymbol{r}^T \boldsymbol{d}} \times \frac{\lambda_{\min}}{2} \|\Delta\|_{\mathrm{T}}^2,$$

where $(a)$ follows because $\|\Delta\|_{1,1,1} \leq \sqrt{k_1 k_2 k_3} \|\Delta\|_{\mathrm{T}}$, $(b)$ follows from (24), $(c)$ follows from the Courant-Fischer theorem (because $\mathbb{E}[\mathrm{vec}(\varPhi(\mathbf{x}))\mathrm{vec}(\varPhi(\mathbf{x}))^T]$ is a symmetric matrix) and Assumption 4.1, and $(d)$ follows because $\|\mathrm{vec}(\Delta)\|_2 = \|\Delta\|_{\mathrm{T}}$. $\qquad\square$

# F   Bounds on the tensor maximum norm of the gradient of the loss function

In this Section, we will show that, with enough samples, the tensor maximum norm of the gradient of the loss function evaluated at the true natural parameter is bounded with high probability. This result will allow us to prove Theorem 4.3 in Appendix G.

We will first state the main result of this Section (Proposition F.1). Next, we will provide a supporting Lemma wherein we show that the expected value of a random variable of interest is zero. Finally, we will prove Proposition F.1.

**Proposition F.1.** *Let Assumptions 2.1, 2.2 and 2.3 be satisfied. For any $\delta_4 \in (0,1)$, any $\epsilon_4 > 0$, the components of the gradient of the loss function $\mathcal{L}_n(\Theta)$[7] evaluated at $\Theta^*$ are bounded from above as*

$$\|\nabla \mathcal{L}_n(\Theta^*)\|_{\max} \leq \epsilon_4,$$

*with probability at least $1 - \delta_4$ as long as*

$$n > \frac{2\phi_{\max}^2 \exp(2\boldsymbol{r}^T \boldsymbol{d})}{\epsilon_4^2} \log\left(\frac{2k_1 k_2 k_3}{\delta_4}\right).$$

## F.1   Supporting Lemma for Proposition F.1

**Lemma F.1.** *For any $u \in [k_1]$, $v \in [k_2]$ and $w \in [k_3]$, define the random variable*

$$x_{uvw} = -\varPhi_{uvw}(\mathbf{x}) \exp\big(-\langle\langle \Theta^*, \varPhi(\mathbf{x})\rangle\rangle\big). \tag{27}$$

---

[7]Ideally, one would consider the gradient of $\mathcal{L}_n(\mathrm{vec}(\Theta))$. However, for the ease of the exposition we abuse the terminology.

*We have*

$$\mathbb{E}[x_{uvw}] = 0,$$

*where the expectation is with respect to $f_{\mathbf{x}}(\mathbf{x}; \Theta^*)$.*

*Proof of Lemma F.1.* Fix any $u \in [k_1]$, $v \in [k_2]$ and $w \in [k_3]$. Using (27), we have

$$\mathbb{E}[x_{uvw}] = -\int_{\mathbf{x} \in \mathcal{X}} f_{\mathbf{x}}(\mathbf{x}; \Theta^*) \Phi_{uvw}(\mathbf{x}) \exp\big(-\langle\langle \Theta^*, \Phi(\mathbf{x})\rangle\rangle\big) d\mathbf{x} \stackrel{(a)}{=} \frac{-\int_{\mathbf{x} \in \mathcal{X}} \Phi_{uvw}(\mathbf{x}) d\mathbf{x}}{\int_{\mathbf{y} \in \mathcal{X}} \exp\big(\langle\langle \Theta^*, \Phi(\mathbf{y})\rangle\rangle\big) d\mathbf{y}}$$

$$\stackrel{(b)}{=} 0,$$

where $(a)$ follows from the definition of $f_{\mathbf{x}}(\mathbf{x}; \Theta^*)$, and because $\mathbb{E}_{\mathcal{U}_{\mathcal{X}}}[\Phi(\mathbf{x})]$ is a constant, and $(b)$ follows because $\int_{\mathbf{x} \in \mathcal{X}} \Phi(\mathbf{x}) d\mathbf{x} = 0$ from Definition 2.1 $\qquad\square$

### F.2   Proof of Proposition F.1

*Proof of Proposition F.1.* Fix $u \in [k_1]$, $v \in [k_2]$ and $w \in [k_3]$. We will start by simplifying the gradient of the $\mathcal{L}_n(\Theta)$ evaluated at $\Theta^*$. The component of the gradient of $\mathcal{L}_n(\Theta)$ corresponding to $\Theta_{uvw}$ evaluated at $\Theta^*$ is given by

$$\frac{\partial \mathcal{L}_n(\Theta^*)}{\partial \Theta_{uvw}} = -\frac{1}{n} \sum_{t=1}^{n} \Phi_{uvw}(\mathbf{x}^{(t)}) \exp\Big(-\Big\langle\Big\langle \Theta^*, \Phi(\mathbf{x}^{(t)})\Big\rangle\Big\rangle\Big).$$

Each term in the above summation is distributed as the random variable $x_{uvw}$ (see (27)). The random variable $x_{uvw}$ has zero mean (see Lemma F.1) and satisfies $|x_{uvw}| \leq \phi_{\max} \exp(\boldsymbol{r}^T \boldsymbol{d})$ (from Assumption 2.3 and (15)). Using the Hoeffding's inequality, we have

$$\mathbb{P}\left(\left|\frac{\partial \mathcal{L}_n(\Theta^*)}{\partial \Theta_{uvw}}\right| > \epsilon_4\right) < 2 \exp\left(-\frac{n\epsilon_4^2}{2\phi_{\max}^2 \exp(2\boldsymbol{r}^T \boldsymbol{d})}\right). \tag{28}$$

The proof follows by using (28) and the union bound over all $u \in [k_1]$, $v \in [k_2]$ and $w \in [k_3]$. $\quad\square$

## G   Proof of Theorem 4.3

In this Section, we will prove Theorem 4.3. We restate the Theorem below and then provide the proof.

**Theorem 4.3.** *Let $\hat{\Theta}_{\epsilon,n}$ be an $\epsilon$-optimal solution of $\hat{\Theta}_n$ obtained from Algorithm 1 for $\epsilon$ of the order $O(\alpha^2 \lambda_{\min})$. Let Assumptions 2.1, 2.2, 2.3, and 4.1 be satisfied. Recall Property 4.1. Then, for any $\delta \in (0, 1)$, we have $\|\hat{\Theta}_{\epsilon,n} - \Theta^*\|_T \leq \alpha$ with probability at least $1 - \delta$ as long as*

$$n \geq O\left(\frac{k_1^2 k_2^2}{\alpha^4 \lambda_{\min}^2} \log\left(\frac{k_1 k_2}{\delta}\right)\right). \tag{13}$$

*The computational cost scales as $O\left(\frac{k_1 k_2}{\alpha^2} \max\left(k_1 k_2 n, c(\Lambda)\right)\right)$ where $c(\Lambda)$ is the cost of projection onto $\Lambda$. Further, ignoring the dependence on $\delta$, $\lambda_{\min}$, and $c(\Lambda)$, $n$ in (13) (as well as the associated computational cost) scales as $O\left(\text{poly}\left(\frac{k_1 k_2}{\alpha}\right)\right)$.*

*Proof of Theorem 4.3.* Let the number of samples satisfy

$$n \geq \max\left\{\frac{8\phi_{\max}^4 k_1^2 k_2^2 k_3^2}{\lambda_{\min}^2} \log\left(\frac{4k_1^2 k_2^2 k_3^2}{\delta}\right),\right.$$

$$\left.\frac{2^9 \phi_{\max}^2 k_1^2 k_2^2 (\boldsymbol{r}^T \boldsymbol{g})^2 (1 + \boldsymbol{r}^T \boldsymbol{d})^2 \exp(4\boldsymbol{r}^T \boldsymbol{d})}{\alpha^4 \lambda_{\min}^2} \log\left(\frac{4k_1 k_2 k_3}{\delta}\right)\right\}$$

$$\stackrel{(a)}{\approx} O\left(\frac{k_1^2 k_2^2}{\alpha^4 \lambda_{\min}^2} \log\left(\frac{k_1 k_2}{\delta}\right)\right) \approx O\left(\text{poly}\left(\frac{k_1 k_2}{\alpha}\right)\right).$$

where $(a)$ follows because $k_3, \phi_{\max}, \boldsymbol{r}, \boldsymbol{g}, \boldsymbol{d} = O(1)$.

Let $\Delta = \hat{\Theta}_{\epsilon,n} - \Theta^*$. Define the residual of the first-order Taylor expansion as

$$\delta\mathcal{L}_n(\Delta, \Theta^*) = \mathcal{L}_n(\Theta^* + \Delta) - \mathcal{L}_n(\Theta^*) - \langle\langle \nabla\mathcal{L}_n(\Theta^*), \Delta \rangle\rangle. \tag{29}$$

Let $\nabla\mathcal{L}_n^{(i)}(\Theta^*)$ denote the $i^{th}$ slice of $\nabla\mathcal{L}_n(\Theta^*)$. From the definition of an $\epsilon$-optimal solution of $\hat{\Theta}_n$, we have

$$\epsilon \geq \mathcal{L}_n(\hat{\Theta}_{\epsilon,n}) - \min_{\Theta \in \Lambda} \mathcal{L}_n(\Theta)$$

$$\geq \mathcal{L}_n(\hat{\Theta}_{\epsilon,n}) - \mathcal{L}_n(\Theta^*)$$

$$\stackrel{(a)}{=} \langle\langle \nabla\mathcal{L}_n(\Theta^*), \hat{\Theta}_{\epsilon,n} - \Theta^* \rangle\rangle + \delta\mathcal{L}_n(\Delta, \Theta^*)$$

$$\stackrel{(b)}{=} \sum_{i=1}^{k_3} \langle \nabla\mathcal{L}_n^{(i)}(\Theta^*), \hat{\Theta}_{\epsilon,n}^{(i)} - \Theta^{*(i)} \rangle + \delta\mathcal{L}_n(\Delta, \Theta^*)$$

$$\stackrel{(c)}{\geq} -\sum_{i=1}^{k_3} \mathcal{R}_i^*(\nabla\mathcal{L}_n^{(i)}(\Theta^*)) \times \mathcal{R}(\hat{\Theta}_{\epsilon,n}^{(i)} - \Theta^{*(i)}) + \delta\mathcal{L}_n(\Delta, \Theta^*)$$

$$\stackrel{(d)}{\geq} -2\sum_{i=1}^{k_3} \mathcal{R}_i^*(\nabla\mathcal{L}_n^{(i)}(\Theta^*)) \times r_i + \delta\mathcal{L}_n(\Delta, \Theta^*)$$

$$\stackrel{(e)}{\geq} -2k_1 k_2 \sum_{i=1}^{k_3} g_i \times \times \|\nabla\mathcal{L}_n^{(i)}(\Theta^*)\|_{\max} \times r_i + \delta\mathcal{L}_n(\Delta, \Theta^*)$$

$$\stackrel{(f)}{\geq} -2k_1 k_2 \|\nabla\mathcal{L}_n(\Theta^*)\|_{\max} \sum_{i=1}^{k_3} g_i \times r_i + \delta\mathcal{L}_n(\Delta, \Theta^*),$$

where $(a)$ follows from (29), $(b)$ follows from the definitions of a slice of a tensor, tensor inner product, and Frobenius inner product, $(c)$ follows from the definition of a dual norm, $(d)$ follows because $\mathcal{R}(\hat{\Theta}_{\epsilon,n}^{(i)} - \Theta^{*(i)}) \leq \mathcal{R}(\hat{\Theta}_{\epsilon,n}^{(i)}) + \mathcal{R}(\Theta^{*(i)}) \leq 2r_i$ from Assumption 2.1, $(e)$ follows from Property 4.1 in Section 4, and $(f)$ follows because $\|\nabla\mathcal{L}_n^{(i)}(\Theta^*)\|_{\max} \leq \|\nabla\mathcal{L}_n(\Theta^*)\|_{\max} \ \forall i \in [k_3]$.

Using Proposition E.1 with $\delta_3 = \frac{\delta}{2}$, and Proposition F.1 with $\delta_4 = \frac{\delta}{2}$, we have the following with probability at least $1 - \delta$.

$$\epsilon \geq -2k_1 k_2 \epsilon_4 \times \boldsymbol{r}^T \boldsymbol{g} + \frac{\lambda_{\min} \exp(-\boldsymbol{r}^T \boldsymbol{d})}{4(1 + \boldsymbol{r}^T \boldsymbol{d})} \|\Delta\|_{\mathrm{T}}^2.$$

This can be rearranged

$$\|\Delta\|_{\mathrm{T}}^2 \leq \frac{\epsilon + 2k_1 k_2 \epsilon_4 \times \boldsymbol{r}^T \boldsymbol{g}}{\lambda_{\min}} \times 4(1 + \boldsymbol{r}^T \boldsymbol{d}) \exp(\boldsymbol{r}^T \boldsymbol{d}). \tag{30}$$

Now, let

$$\epsilon = \frac{\alpha^2 \lambda_{\min}}{8(1 + \boldsymbol{r}^T \boldsymbol{d}) \exp(\boldsymbol{r}^T \boldsymbol{d})} \quad \text{and} \quad \epsilon_4 = \frac{\alpha^2 \lambda_{\min}}{16 k_1 k_2 \times \boldsymbol{r}^T \boldsymbol{g} \times (1 + \boldsymbol{r}^T \boldsymbol{d}) \times \exp(\boldsymbol{r}^T \boldsymbol{d})}. \tag{31}$$

Plugging in $\epsilon$ and $\epsilon_4$ from (31) in (30), we obtain that

$$\|\Delta\|_{\mathrm{T}} \leq \alpha.$$

The computational cost of the operation $\Theta_{(t)} - \eta\nabla\mathcal{L}_n(\Theta_{(t)}) - \Theta$ in Algorithm 1 is of the order $k_1 k_2 n$ (because $k_3 = O(1)$). Therefore, the computational cost of the step $\Theta_{(t+1)} \leftarrow \arg\min_{\Theta \in \Lambda} \|\Theta_{(t)} - \eta\nabla\mathcal{L}_n(\Theta_{(t)}) - \Theta\|$ of Algorithm 1 is of the order $\max\{k_1 k_2 n, c(\Lambda)\}$. From Lemma 3.1, with $\epsilon = O(\alpha^2 \lambda_{\min})$, Algorithm 1 returns an $\epsilon$-optimal solution $\hat{\Theta}_{\epsilon,n}$ as long as $\tau = O\left(\text{poly}\left(\frac{k_1 k_2}{\alpha^2 \lambda_{\min}}\right)\right)$.

Therefore, the total computational cost scales as $O\left(\frac{k_1 k_2}{\alpha^2 \lambda_{\min}} \max\left(k_1 k_2 n, c(\Lambda)\right)\right)$. Whenever the cost of projection onto $\Lambda$ is $O(\text{poly}(k_1 k_2))$, we have the total computational cost scaling as $O\left(\text{poly}\left(\frac{k_1 k_2}{\alpha}\right)\right)$. □

# H  Computational cost for the example constraints on the natural parameters

In this Section, we provide Corollary H.1, Corollary H.2, and Corollary H.3. These Corollaries provide the computational cost to produce an $\epsilon$-optimal solution of $\hat{\Theta}_n$ for sparse decomposition of $\Theta$, low-rank decomposition of $\Theta$, and sparse-plus-low-rank decomposition of $\Theta$. respectively. Recall the convex relaxations of these constraints from Section 2.1.

## H.1  Sparse Decomposition

**Corollary H.1.** *(Sparse decomposition) Suppose $\Theta^*$ has a sparse decomposition i.e., $\Theta^* = (\Theta^{*(1)})$ and $\|\Theta^{*(1)}\|_{1,1} \leq r_1$. Let Assumptions 2.1, 2.2, 2.3, and 4.1 be satisfied. Let*

$$n \geq O\left(\frac{k_1^2 k_2^2}{\alpha^4 \lambda_{\min}^2} \log\left(\frac{k_1 k_2}{\delta}\right)\right).$$

*Let $\eta = 1/k_1 k_2 k_3 \phi_{\max}^2 \exp(r_1 d_1)$ and $\Theta^{(0)} = \mathbf{0}$. Then, Algorithm 1 is guaranteed to produce an $\epsilon$-optimal solution $\hat{\Theta}_{\epsilon,n}$ such that $\|\hat{\Theta}_{\epsilon,n} - \Theta^*\|_{\mathrm{T}} \leq \alpha$, with probability at least $1 - \delta$ and with number of computations of the order*

$$O\left(\frac{k_1^4 k_2^4}{\alpha^6 \lambda_{\min}^3} \log\left(\frac{k_1 k_2}{\delta}\right)\right).$$

*Proof of Corollary H.1 .*  The computational cost of projecting on the $L_{1,1}$ ball is $O(k_1 k_2)$ (see [15] and note $k_3 = O(1)$). The computational cost of the operation $\Theta_{(t)} - \eta \nabla \mathcal{L}_n(\Theta_{(t)}) - \Theta$ is $O(k_1 k_2 n)$ (because $k_3 = O(1)$). Therefore, the computational cost of the step $\Theta_{(t+1)} \leftarrow \arg\min_{\Theta \in \Lambda} \|\Theta_{(t)} - \eta \nabla \mathcal{L}_n(\Theta_{(t)}) - \Theta\|$ of Algorithm 1 is $O(k_1 k_2 n)$.

From Lemma 3.1, Algorithm 1 returns an $\epsilon$-optimal solution $\hat{\Theta}_{\epsilon,n}$ as long as

$$\tau \geq \frac{2 k_1 k_2 \phi_{\max}^2 \exp(\boldsymbol{r}^T \boldsymbol{d})}{\epsilon} \|\hat{\Theta}_n\|_{\mathrm{T}}^2.$$

Also, $\|\hat{\Theta}_n\|_{\mathrm{T}}^2 = \|\hat{\Theta}_n^{(1)}\|_{\mathrm{F}}^2 \leq \|\hat{\Theta}_n^{(1)}\|_{1,1}^2 \leq r_1^2$. Combining everything, the computational cost scales as $O\left(\frac{k_1^2 k_2^2 n}{\epsilon}\right)$. Using Theorem 4.3, and plugging in $n = O\left(\frac{k_1^2 k_2^2}{\alpha^4 \lambda_{\min}^2} \log\left(\frac{k_1 k_2}{\delta}\right)\right)$ and $\epsilon = O(\alpha^2 \lambda_{\min})$ completes the proof. $\qquad\square$

## H.2  Low-rank decomposition

**Corollary H.2.** *(Low-rank decomposition) Suppose $\Theta^*$ has a low-rank decomposition i.e., $\Theta^* = (\Theta^{*(1)})$ and $\|\Theta^*\|_\star \leq r_1$. Let Assumptions 2.1, 2.2, 2.3, and 4.1 be satisfied. Let*

$$n \geq O\left(\frac{k_1^2 k_2^2}{\alpha^4 \lambda_{\min}^2} \log\left(\frac{k_1 k_2}{\delta}\right)\right).$$

*Let $\eta = 1/k_1 k_2 k_3 \phi_{\max}^2 \exp(r_1 d_1)$ and $\Theta^{(0)} = \mathbf{0}$. Then, Algorithm 1 is guaranteed to produce an $\epsilon$-optimal solution $\hat{\Theta}_{\epsilon,n}$ such that $\|\hat{\Theta}_{\epsilon,n} - \Theta^*\|_{\mathrm{T}} \leq \alpha$, with probability at least $1 - \delta$ and with number of computations of the order*

$$O\left(\frac{k_1^4 k_2^4}{\alpha^6 \lambda_{\min}^3} \log\left(\frac{k_1 k_2}{\delta}\right)\right).$$

*Proof of Corollary H.2 .*  The computational cost of projecting on the nuclear ball is $O(k_1 k_2 \min\{k_1, k_2\})$ (see [23] and note $k_3 = O(1)$). The computational cost of the operation $\Theta_{(t)} - \eta \nabla \mathcal{L}_n(\Theta_{(t)}) - \Theta$ is $O(k_1 k_2 n)$ because $(k_3 = O(1))$. Therefore, the computational cost of the step $\Theta_{(t+1)} \leftarrow \arg\min_{\Theta \in \Lambda} \|\Theta_{(t)} - \eta \nabla \mathcal{L}_n(\Theta_{(t)}) - \Theta\|$ of Algorithm 1 is $O(k_1 k_2 \max\{\min\{k_1, k_2\}, n\})$.

From Lemma 3.1, Algorithm 1 returns an $\epsilon$-optimal solution $\hat{\Theta}_{\epsilon,n}$ scales as

$$\tau \geq \frac{2k_1 k_2 \phi_{\max}^2 \exp(\boldsymbol{r}^T \boldsymbol{d})}{\epsilon} \|\hat{\Theta}_n\|_{\mathrm{F}}^2.$$

Also, $\|\hat{\Theta}_n\|_{\mathrm{F}}^2 \leq \|\hat{\Theta}_n\|_{\star}^2 \leq r_1^2$. Combining everything, the computational cost is of the order $O\left(\frac{k_1^2 k_2^2 \max\{\min\{k_1,k_2\},n\}}{\epsilon}\right)$. Using Theorem 4.3, and plugging in $n = O\left(\frac{k_1^2 k_2^2}{\alpha^4 \lambda_{\min}^2} \log\left(\frac{k_1 k_2}{\delta}\right)\right)$ and $\epsilon = O(\alpha^2 \lambda_{\min})$ completes the proof. $\qquad\square$

### H.3   Sparse-plus-low-rank decomposition

**Corollary H.3.** *(Sparse-plus-low-rank decomposition) Suppose $\Theta^*$ has a sparse-plus-low-rank decomposition i.e., $\Theta^* = (\Theta^{*(1)}, \Theta^{*(2)})$ such that $\|\Theta^{*(1)}\|_{1,1} \leq r_1$ and $\|\Theta^{*(2)}\|_{\star} \leq r_2$. Let Assumptions 2.1, 2.2, 2.3, and 4.1 be satisfied. Let*

$$n \geq O\left(\frac{k_1^2 k_2^2}{\alpha^4 \lambda_{\min}^2} \log\left(\frac{k_1 k_2}{\delta}\right)\right).$$

*Let $\eta = 1/k_1 k_2 k_3 \phi_{\max}^2 \exp(r_1 d_1 + r_2 d_2)$ and $\Theta^{(0)} = \mathbf{0}$. Then, Algorithm 1 is guaranteed to produce an $\epsilon$-optimal solution $\hat{\Theta}_{\epsilon,n}$ such that $\|\hat{\Theta}_{\epsilon,n} - \Theta^*\|_{\mathrm{T}} \leq \alpha$, with probability at least $1 - \delta$ and with number of computations of the order*

$$O\left(\frac{k_1^4 k_2^4}{\alpha^6 \lambda_{\min}^3} \log\left(\frac{k_1 k_2}{\delta}\right)\right).$$

*Proof of Corollary H.3 .* The proof follows directly from the proofs of Corollary H.1 and Corollary H.2. $\qquad\square$

## I   Examples

In this Section, we provide a more elaborate discussion on the examples of natural parameters and statistics from Section 2.1.

### I.1   Sparse-plus-low-rank decomposition

The natural statistic $\Phi$ of an exponential family is such that for any $i_1 \neq i_2 \in [k_1], j_1 \neq j_2 \in [k_2], l_1 \neq l_2 \in [k_3]$, $\Phi_{i_1 j_1 l_1} \neq \Phi_{i_2 j_2 l_2}$. Further, an exponential family is minimal if there does not exist a non-zero tensor $\mathbf{U} \in \mathbb{R}^{k_1 \times k_2 \times k_3}$ such that $\sum_{i \in [k_1], j \in [k_2], l \in [k_3]} \mathbf{U}_{ijl} \Phi_{ijl}(\mathbf{x})$ is equal to a constant for all $\mathbf{x} \in \mathcal{X}$. However, for the sparse-plus-low-rank decomposition, it is desirable to let $\Phi^{(1)} = \Phi^{(2)}$ (see [8, 37]). In this scenario, there exists a non-zero tensor $\mathbf{U} \in \mathbb{R}^{k_1 \times k_2 \times k_3}$ such that $\sum_{i \in [k_1], j \in [k_2], l \in [k_3]} \mathbf{U}_{ijl} \Phi_{ijl}(\mathbf{x}) = 0$ for all $\mathbf{x} \in \mathcal{X}$ for e.g., this is true if $\mathbf{U}^{(1)} = -\mathbf{U}^{(2)}$. In this situation, we say an exponential family is minimal if there does not exist a non-zero tensor $\mathbf{U} \in \mathbb{R}^{k_1 \times k_2 \times k_3}$ such that $\sum_{l \in [k_3]} \mathbf{U}^{(l)} \neq 0$ as well as $\sum_{i \in [k_1], j \in [k_2], l \in [k_3]} \mathbf{U}_{ijl} \Phi_{ijl}(\mathbf{x})$ is equal to a constant for all $\mathbf{x} \in \mathcal{X}$. Therefore, it is often convenient to represent the tensor $\mathbf{U}$ in terms of a matrix and define minimality of an exponential family in terms of this new matrix.

### I.2   Assumptions 2.1 and 2.2

While we expect the constants $\boldsymbol{r}$ in Assumption 2.1 and $\boldsymbol{d}$ in Assumption 2.2 to be $O(1)$ for most applications, the sample complexity and the computational complexity in Theorem 4.3 would still be $O\left(\mathrm{poly}\left(\frac{k_1 k_2}{\alpha}\right)\right)$ as long as $\boldsymbol{r}$ and $\boldsymbol{d}$ are $O\left(\log(k_1 k_2)\right)$.

### I.3   Polynomial natural statistic

Suppose the natural statistics are polynomials of $\mathbf{x}$ with maximum degree $l$, i.e., $\prod_{i \in [p]} x_i^{l_i}$ such that $l_i \geq 0 \; \forall i \in [p]$ and $\sum_{i \in [p]} l_i \leq l$.

- Let $\mathcal{X} = [0, b]$ for $b \in \mathbb{R}$. We will first show that $\phi_{\max} = 2b^l$. We have

$$
\begin{aligned}
\|\varPhi(\mathbf{x})\|_{\max} &= \max_{u \in [k_1], v \in [k_2], w \in [k_3]} |\varPhi_{uvw}(\mathbf{x})| \\
&\overset{(a)}{=} \max_{u \in [k_1], v \in [k_2], w \in [k_3]} \left| \varPhi_{uvw}(\mathbf{x}) - \mathbb{E}_{\mathcal{U}_{\mathcal{X}}}[\varPhi_{uvw}(\mathbf{x})] \right| \\
&\overset{(b)}{\leq} \max_{u \in [k_1], v \in [k_2], w \in [k_3]} \left| \varPhi_{uvw}(\mathbf{x}) \right| + \max_{u \in [k_1], v \in [k_2], w \in [k_3]} \left| \mathbb{E}_{\mathcal{U}_{\mathcal{X}}}[\varPhi_{uvw}(\mathbf{x})] \right| \\
&\leq 2 \max_{\mathbf{x} \in \mathcal{X}} \max_{u \in [k_1], v \in [k_2], w \in [k_3]} \left| \varPhi_{uvw}(\mathbf{x}) \right| \leq 2b^l.
\end{aligned}
$$

  where $(a)$ follows from Definition 2.1 and $(b)$ follows from the triangle inequality.
- Suppose $\Theta^*$ has a sparse decomposition i.e., $\Theta^* = (\Theta^{*(1)})$ and $\|\Theta^{*(1)}\|_{1,1} \leq r_1$. The dual norm of the matrix $L_{1,1}$ norm is the matrix maximum norm. Then, if $\mathcal{X} = [0, b]$ for $b \in \mathbb{R}$,

$$
\mathcal{R}_1^*(\varPhi^{(1)}(\mathbf{x})) = \|\varPhi^{(1)}(\mathbf{x})\|_{\max} = \|\varPhi(\mathbf{x})\|_{\max} \leq \phi_{\max} = 2b^l.
$$

- Suppose $\Theta^*$ has a low-rank decomposition i.e., $\Theta^* = (\Theta^{*(1)})$ and $\|\Theta^*\|_\star \leq r_1$. The dual norm of the matrix nuclear norm is the matrix spectral norm. Then,

$$
\mathcal{R}_1^*(\varPhi^{(1)}(\mathbf{x})) = \|\varPhi^{(1)}(\mathbf{x})\|.
$$

  Let $l = 2$, and $\mathcal{X} = \mathcal{B}(0, b)$. Observe that by writing $\varPhi^{(1)}(\mathbf{x}) = \tilde{x}\tilde{x}^T$ where $\tilde{x} = (1, x_1, \cdots, x_p)$, we have

$$
\|\varPhi^{(1)}(\mathbf{x})\| \leq 2\Big(1 + \sum_{i \in [p]} \mathbf{x}_i^2 \Big) \leq 2(1 + b^2).
$$

- Suppose $\Theta^*$ has a sparse-plus-low-rank decomposition i.e., $\Theta^* = (\Theta^{*(1)}, \Theta^{*(2)})$ such that $\|\Theta^{*(1)}\|_{1,1} \leq r_1$ and $\|\Theta^{*(2)}\|_\star \leq r_2$. The dual norm of the matrix $L_{1,1}$ norm is the matrix maximum norm and the dual norm of the matrix nuclear norm is the matrix spectral norm. Let $l = 2$, and $\mathcal{X} = \mathcal{B}(0, b)$. Then,

$$
\mathcal{R}^*(\varPhi(\mathbf{x})) \leq (\|\varPhi^{(1)}(\mathbf{x})\|_{\max}, \|\varPhi^{(2)}(\mathbf{x})\|) \leq (2b^2, 2 + 2b^2).
$$

## I.4   Trigonometric natural statistic

Suppose the natural statistics are sines and cosines of $\mathbf{x}$ with $l$ different frequencies, i.e., $\sin(\sum_{i \in [p]} l_i x_i) \cup \cos(\sum_{i \in [p]} l_i x_i)$ such that $l_i \in [l] \cup \{0\}$.

- Let $\mathcal{X} \subset \mathbb{R}^p$. We will first show that $\phi_{\max} = 2$. We have

$$
\begin{aligned}
\|\varPhi(\mathbf{x})\|_{\max} &= \max_{u \in [k_1], v \in [k_2], w \in [k_3]} |\varPhi_{uvw}(\mathbf{x})| \\
&\overset{(a)}{=} \max_{u \in [k_1], v \in [k_2], w \in [k_3]} \left| \varPhi_{uvw}(\mathbf{x}) - \mathbb{E}_{\mathcal{U}_{\mathcal{X}}}[\varPhi_{uvw}(\mathbf{x})] \right| \\
&\overset{(b)}{\leq} \max_{u \in [k_1], v \in [k_2], w \in [k_3]} \left| \varPhi_{uvw}(\mathbf{x}) \right| + \max_{u \in [k_1], v \in [k_2], w \in [k_3]} \left| \mathbb{E}_{\mathcal{U}_{\mathcal{X}}}[\varPhi_{uvw}(\mathbf{x})] \right| \\
&\leq 2 \max_{\mathbf{x} \in \mathcal{X}} \max_{u \in [k_1], v \in [k_2], w \in [k_3]} \left| \varPhi_{uvw}(\mathbf{x}) \right| \leq 2.
\end{aligned}
$$

  where $(a)$ follows from Definition 2.1 and $(b)$ follows from the triangle inequality.
- Suppose $\Theta^*$ has a sparse decomposition i.e., $\Theta^* = (\Theta^{*(1)})$ and $\|\Theta^{*(1)}\|_{1,1} \leq r_1$. The dual norm of the matrix $L_{1,1}$ norm is the matrix maximum norm. Then, for any $\mathcal{X} \subset \mathbb{R}^p$,

$$
\mathcal{R}_1^*(\varPhi^{(1)}(\mathbf{x})) = \|\varPhi^{(1)}(\mathbf{x})\|_{\max} = \|\varPhi(\mathbf{x})\|_{\max} \leq \phi_{\max} = 2.
$$

## I.5 Combinations of polynomial and trigonometric statistics

Suppose the natural statistics are combinations of polynomials of $\mathbf{x}$ with maximum degree $l$, i.e., $\prod_{i\in[p]} x_i^{l_i}$ such that $l_i \geq 0 \;\forall i \in [p]$ and $\sum_{i\in[p]} l_i \leq l$ as well as sines and cosines of $\mathbf{x}$ with $\tilde{l}$ different frequencies, i.e., $\sin(\sum_{i\in[p]} l_i x_i) \cup \cos(\sum_{i\in[p]} l_i x_i)$ such that $l_i \in [\tilde{l}] \cup \{0\}$.

- Let $\mathcal{X} = [0, b]$ for $b \in \mathbb{R}$. From Appendix I.3 and Appendix I.4, it is easy to verify that $\phi_{\max} = \max\{2, 2b^l\}$.
- Suppose $\Theta^*$ has a sparse decomposition i.e., $\Theta^* = (\Theta^{*(1)})$ and $\|\Theta^{*(1)}\|_{1,1} \leq r_1$. The dual norm of the matrix $L_{1,1}$ norm is the matrix maximum norm. Then, if $\mathcal{X} = [0, b]$ for $b \in \mathbb{R}$, it is easy to verify that

$$\mathcal{R}_1^*(\Phi^{(1)}(\mathbf{x})) = \|\Phi^{(1)}(\mathbf{x})\|_{\max} = \|\Phi(\mathbf{x})\|_{\max} \leq \phi_{\max} = \max\{2, 2b^l\}.$$

# J  Property 4.1 for norms of interest

In this Section, we show that the $g$ defined in Property 4.1 in Section 4 is 1 for the entry-wise $L_{p,q}$ norm $(p, q \geq 1)$, the Schatten $p$-norm $(p \geq 1)$, and the operator $p$-norm $(p \geq 1)$.

## J.1  The entry-wise $L_{p,q}$ norm

Let $\tilde{\mathcal{R}}(\cdot)$ denote the entry-wise $L_{p,q}$ norm for some $p, q \geq 1$. We will show that for any matrix $\mathbf{M} \in \mathbb{R}^{k_1 \times k_2}$

$$\tilde{\mathcal{R}}(\mathbf{M}) \leq \|\mathbf{M}\|_{\max} \times k_1^{\frac{1}{p}} k_2^{\frac{1}{q}}.$$

By the definition of the entry-wise $L_{p,q}$ norm, we have

$$\tilde{\mathcal{R}}(\mathbf{M}) = \left( \sum_{j\in[k_2]} \left( \sum_{i\in[k_1]} |M_{ij}|^p \right)^{\frac{q}{p}} \right)^{\frac{1}{q}} \leq \left( \sum_{j\in[k_2]} \left( \sum_{i\in[k_1]} \|\mathbf{M}\|_{\max}^p \right)^{\frac{q}{p}} \right)^{\frac{1}{q}}$$

$$= k_1^{\frac{1}{p}} k_2^{\frac{1}{q}} \|\mathbf{M}\|_{\max} \leq k_1 k_2 \|\mathbf{M}\|_{\max}.$$

## J.2  The Schatten $p$-norm

Let $\tilde{\mathcal{R}}(\cdot)$ denote the Schatten $p$-norm for some $p \geq 1$. We will show that for any matrix $\mathbf{M} \in \mathbb{R}^{k_1 \times k_2}$

$$\tilde{\mathcal{R}}(\mathbf{M}) \leq \|\mathbf{M}\|_{\max} \times \sqrt{\min\{k_1, k_2\} k_1 k_2}.$$

Let the rank of $\mathbf{M}$ be denoted by $r$ and the singular values of $\mathbf{M}$ be denoted by $\sigma_i(\mathbf{M})$ for $i \in [r]$. By the definition of the Schatten $p$-norm, we have

$$\tilde{\mathcal{R}}(\mathbf{M}) = \left( \sum_{i\in[r]} \sigma_i^p(\mathbf{M}) \right)^{\frac{1}{p}} \overset{(a)}{\leq} \sum_{i\in[r]} \sigma_i(\mathbf{M}) \overset{(b)}{\leq} \sqrt{r k_1 k_2} \|\mathbf{M}\|_{\max}$$

$$\overset{(c)}{\leq} \sqrt{\min\{k_1, k_2\} k_1 k_2} \|\mathbf{M}\|_{\max} \leq k_1 k_2 \|\mathbf{M}\|_{\max}$$

where $(a)$ follows because of the monotonicity of the Schatten $p$-norms, $(b)$ follows because $\|\mathbf{M}\|_\star \leq \sqrt{r k_1 k_2} \|\mathbf{M}\|_{\max}$, and $(c)$ follows because $r \leq \min\{k_1, k_2\}$.

## J.3  The operator $p$-norm

Let $\tilde{\mathcal{R}}(\cdot)$ denote the operator $p$-norm for some $p \geq 1$. We will show that for any matrix $\mathbf{M} \in \mathbb{R}^{k_1 \times k_2}$

$$\tilde{\mathcal{R}}(\mathbf{M}) \leq \|\mathbf{M}\|_{\max} \times k_1^{\frac{1}{p}} k_2^{1 - \frac{1}{p}}.$$

Let $q = \frac{p}{p-1}$. For $i \in k_1$, let $[\mathbf{M}]_i$ denote the $i^{th}$ row of $\mathbf{M}$. By the definition of the operator $p$-norm, we have

$$\tilde{\mathcal{R}}(\mathbf{M}) = \max_{\mathbf{y}:\|\mathbf{y}\|_p=1} \|\mathbf{M}\mathbf{y}\|_p \overset{(a)}{\leq} k_1^{\frac{1}{p}} \max_{\mathbf{y}:\|\mathbf{y}\|_p=1} \|\mathbf{M}\mathbf{y}\|_\infty$$

$$\overset{(b)}{\leq} k_1^{\frac{1}{p}} \max_{\mathbf{y}:\|\mathbf{y}\|_p=1} \max_{i \in [k_1]} \|[\mathbf{M}]_i\|_q \|\mathbf{y}\|_p$$

$$\leq k_1^{\frac{1}{p}} \max_{i \in [k_1]} \|[\mathbf{M}]_i\|_q$$

$$\overset{(c)}{\leq} k_1^{\frac{1}{p}} k_2^{\frac{1}{q}} \max_{i \in [k_1]} \|[\mathbf{M}]_i\|_\infty$$

$$= k_1^{\frac{1}{p}} k_2^{1-\frac{1}{p}} \|\mathbf{M}\|_{\max} \leq k_1 k_2 \|\mathbf{M}\|_{\max}$$

where $(a)$ follows because $\|\mathbf{v}\|_p \leq m^{\frac{1}{p}}\|\mathbf{v}\|_\infty$ for any vector $\mathbf{v} \in \mathbb{R}^m$ and $p \geq 1$, $(b)$ follows from the definition of the infinity norm of a vector and using the Hölder's inequality, and $(c)$ follows because $\|\mathbf{v}\|_q \leq m^{\frac{1}{q}}\|\mathbf{v}\|_\infty$ for any vector $\mathbf{v} \in \mathbb{R}^m$ and $q \geq 1$.