# OpenReview forum: "A Computationally Efficient Method for Learning Exponential Family Distributions"
_NeurIPS.cc/2021/Conference — NeurIPS 2021 Poster_

### Official Review · Reviewer_kMiC · 2021-07-02

**Rating:** 5
**Confidence:** 3

**Summary:**

This paper considers the parameter estimation of the minimal exponential family and proposed an estimator by minimizing a convex objective function. The resulting estimate is proved to be consistent, asymptotically normal, and has polynomial sample complexity.

**Limitations And Societal Impact:**

The limitations are discussed in the paper. The potential negative societal impacts are not available as the paper is mostly on the theoretical aspects.

**Main Review:**

The problem considered is of broad interest, and the overall analysis and presentation are clear. The main issue of this paper is the validity of assumptions. The key assumption 2.1 assuming constant r and d needs justification since the dependency on r and d is exponential. Currently, I don't see any other benefits of using the tensor formulation, and this assumption looks pretty restrictive. Also, the author comments on the restriction of boundedness assumption, so it'd be helpful to extract the explicit dependency on the bound. For the results on such a specific problem with a series of assumptions, the notion of poly is not sufficient (btw, it is inappropriate to use $\approx$ in a theorem statement). For instance, the rate $1/\alpha^4$ should be properly discussed.

======

Elaborations on the review comments:
- Tensor formulation: as the authors pointed out, the formulation in (1) and (2) are equivalent. Can you explain the necessity of using tensor formalism? From my understanding, Assumption 2.1 restricts the norm of each slice of the tensor, so it appears that a matrix formulation suffices for that purpose.
- Notion of poly: as the problem is tailored to a parametric formulation, I'd expect some parametric rate. However, the current $1/\alpha^4$ is far from the parametric rate. So please explain more about your result. For instance, can you discuss whether $1/\alpha^4$ is tight or loose?

**Time Spent Reviewing:**

5

---

> ### Author Response · Authors · 2021-08-10
> **Explanation regarding validity of the assumptions; Rate in-line with prior work; No known lower bounds**
>
> We thank Reviewer kMiC  for their detailed feedback and suggestions.
>
> We would like to emphasize that, to be the best of our knowledge (and as pointed out by reviewer kDGo), our work is the first to focus learning of general exponential families in a computationally and statistically efficient manner. To that end, we propose a novel efficient estimator and a practical algorithm (Algorithm 1).
>
> We respond to the reviewer comments below.
>
> 1. _Validity of the assumptions_:
> * We look at Assumption 2.1 as a potential flexibility in the problem specification. In other words, a practitioner has the option to choose from a variety of constraints on the natural parameters (that could be handled by our framework). For example, in some real-world applications the parameters are sparse while in some other real-world applications the parameters have a low-rank and a practitioner could _choose_ either depending on the application at hand.  Focusing on the constant $r$, we expect this to be $O(1)$ for most applications. Even if this is not true, the sample complexity in Theorem 4.3 would still be $O(\sf{poly} (k_1k_2))$ as long as the constant $r$ is  $O(\sf{log} (k_1k_2))$.
> * We believe that for polynomial and/or sinusoidal natural statistics, Assumptions 2.2 and 2.3 would hold whenever the domain of the random vector $\mathbf{x}$ is appropriately bounded. (Aside: Most of the previous work work only with polynomial natural statistics). We provide a variety of examples in section 2.1 that satisfy Assumptions 2.1, 2.2, and 2.3. Focusing on the constant $d$, the sample complexity in Theorem 4.3 would still be $O(\sf{poly} (k_1k_2))$ as long as the constant $d$ is  $O(\sf{log} (k_1k_2))$ (this should be true for most applications with bounded domain).
>
> 2. _Tensor formulation_: We agree with the reviewer that matrix formulation suffices and would not change the analysis. We prefer a 3-way tensor decomposition for notational ease. One could also consider $k_3$ different matrices of dimension $k_1 \times k_2$ instead of a tensor of dimension $k_1 \times k_2 \times k_3$. In summary, the tensor formulation is preferred for a notational convenience.
>
> 3. _Notion of poly, etc_: We will revise our paper showing explicit dependence of the boundness assumption in the sample complexity. Also, we will replace the use of $\approx$ with the more precise statement clarifying the meaning of this approximation. The dependence of $1/\alpha^4$ is in-line with some of other prior works (e.g., [40], [51]) that use a similar loss function. It seems that this dependence on $\alpha$ is fundamental to loss function. There is a brief discussion on this in Appendix E.1 of [51]. We do know that for _sparse_ graphical models (e.g., Ising models), one could achieve a better dependence of $1/\alpha^2$ (see [A, B]). However, it is not yet clear what the lower bound on the sample complexity would be for the general class of exponential families considered in this work (which may not be sparse) or how this lower bound depends on $\alpha$. We believe this is an important question and worth pursuing.
>
> [A] ''Learning graphical models using multiplicative weights'' by Klivans et al. (2017)
>
> [B] ''Interaction screening: Efficient and sample-optimal learning of ising models'' by Vuffray et al. (2016)

---

### Official Review · Reviewer_kDGo · 2021-07-16

**Rating:** 6
**Confidence:** 4

**Summary:**

The authors consider the problem of estimating parameters of an exponential family from i.i.d samples. Although MLE has all the nice statistical guarantees, it is open to efficiently compute it. In this work, the authors overcome the computational difficulty by proposing a new estimator. This new estimator can be computed efficiently and under certain conditions also has nice statistical guarantees except for the asymptotical efficiency.

**Limitations And Societal Impact:**

The authors do a good job of addressing the limitations of their work. Although the current work is theoretical in nature, I would urge authors to think about the potential societal impact of their work.

**Main Review:**

The things that I like about the paper.
1) The problem considered here is fundamental and the results presented are quite interesting.
2) The authors provide the first efficient estimator with nice statistical guarantees. In particular, their estimator achieves polynomial sample complexity.
3) The authors do a good job of discussing the limitation of their work.
4) The paper is well structured and does a clear presentation of the results.

Few concerns/suggestions:
1) Notation was a little confusing at few places. For instance, using italic and bold characters to denote different quantities was difficult to notice. However, I don't have any better suggestions.
2) Maybe include some experiments comparing the performance of the proposed algorithm with previous work on learning Gaussian distributions. I am curious to see the empirical performance of the proposed estimator if the underlying distribution is Gaussian.
3) The proposed estimator doesn't achieve asymptotic efficiency and they pose this as an open problem.
4) As this is the first paper to introduce the problem of estimating parameters of the exponential family. It would have been nice if the authors included some non-trivial information-theoretic lower bounds. Also in the related work, talk about the sample complexity for the Gaussian case and compare it with the more general exponential family.
5) Provide an explanation on why various assumptions in the paper are reasonable/necessary and the difficulties in designing algorithms without these assumptions. In particular, try to provide real-world applications where the assumptions considered in this paper hold.

I would be happy if authors could handle a subset of these suggestions/concerns.

**Time Spent Reviewing:**

8

---

> ### Author Response · Authors · 2021-08-10
> **Explanation regarding the assumptions; Interesting suggestions for future work**
>
> We thank Reviewer kDGo  for their detailed feedback and suggestions. We respond to their concerns/suggestions below.
>
> 1. We thank the reviewer for bringing this to our attention. Even though we tried to be consistent with the notation throughout the paper, we acknowledge that it can be confusing. We are happy to revise the notations in any way the reviewer thinks would help.
>
> 2. We agree with the reviewer that an empirical study of the proposed method will help understand its practicality and contrast it with previous works. Building on our current work, we plan to conduct a detailed empirical exploration in subsequent work.
>
> 3. We thank the reviewer for bringing this up. As mentioned in the limitations section, while our estimator is computationally efficient, consistent, and asymptotically normal, it is not asymptotically efficient. We believe investigating the possibility of a single estimator that achieves computational and asymptotic efficiency for this class of exponential family could be an interesting future direction.
>
> 4. We thank the reviewer for their suggestion. We agree deriving non-trivial lower bounds on the sample complexity for the class of exponential families considered in this work is an important question and worth pursuing. We will revise our paper to include a discussion regarding the sample complexity for the truncated Gaussian case ($d^2/\alpha^2$) and a comparison with the general exponential family ($d^2/\alpha^4$) considered in this work.
>
> 5. We make the following assumptions in our work - Assumption 2.1, 2.2, 2.3, and 4.1.
>
> * A2.1 -  We look at Assumption 2.1 as a potential flexibility in the problem specification. In other words, a practitioner has the option to choose from a variety of constraints on the natural parameters (that could be handled by our framework). For example, in some real-world applications the parameters are sparse while in some other real-world applications the parameters have a low-rank and a practitioner could _choose_ either depending on the application at hand.
>
> * A:2.2+A:2.3 - We believe that for polynomial and/or sinusoidal natural statistics, Assumptions 2.2 and 2.3 would hold whenever the domain of the random vector $\mathbf{x}$ is appropriately bounded. (Aside: Most of the previous work work only with polynoimal natural statistics). We provide a variety of examples in section 2.1 that satisfy Assumptions 2.1, 2.2, and 2.3.
>
> * A:4.1 - We view Assumption 4.1 as an adequate condition to rule out certain singular distributions (It can be seen in the proof of Proposition E.1 that this condition effectively lower bounds the variance of a non-constant random variable). Therefore, we expect this assumption to hold for most real-world applications.
>
>
> 5. (continued) We will add the above discussions in the revision.

---

### Official Review · Reviewer_sAxR · 2021-07-18

**Rating:** 6
**Confidence:** 3

**Summary:**

The paper presents a new estimator of the natural parameters of an exponential family with bounded support. To derive the proposed estimator, the authors first present a new loss function which does not involve the normalizing constant. Then the proposed estimator is defined as the minimizer of the loss function. Consistency and asymptotic normality of the proposed estimator is shown. Also, the condition on the number of samples and that on the computational cost are given to achieve a certain approximation of the proposed estimator to the true parameter.

**Ethical Concerns:**

I do not find any ethical issues with this paper.

**Limitations And Societal Impact:**

The authors have adequately addressed the limitations of their work. As pointed out by the authors in the last section of the paper and by this reviewer in the last paragraph of "Main Review" Section, empirical study is an important direction. I suggest that the authors conduct (at least simple) experiments in this work rather than in future work.

I do not find any potential negative societal impact of this work.

**Main Review:**

Originality: The presented estimator seems new. Theorem 4.1 implies that the population version of the proposed estimator is equivalent to the maximum likelihood estimator (MLE), but its sample version is different in general and is claimed to be calculated more efficiently than the MLE. Related works are well explained and the background of this work is clear.

Quality: The submission seems technically sound. The claims are well supported theoretically. No experimental study is given in the paper.

Clarity: The paper is clearly written in general. Appendix is very helpful because it provides detailed proofs of the theoretical results given in the paper.

Significance: I reckon that the theoretical results of the paper are nice. The presented estimator has two desirable properties, namely, consistency and asymptotic normality. Also, the calculation of the loss function (10) is simpler than the likelihood function in the sense that the proposed loss function does not involve the calculation of the normalizing constant at each step of the algorithm.

However the authors do not conduct any experiment to compare the proposed estimator with MLE and other existing estimators in terms of their performance and computational cost. Without the experiments, it is difficult to convince readers of the importance of the proposed estimator. Although Theorem 4.3 is helpful to evaluate the performance and computational cost of the proposed estimator, I think empirical results are necessary to verify the theoretical results.

**Time Spent Reviewing:**

5

---

> ### Author Response · Authors · 2021-08-10
> **Emphasize on being the first work to theoretically analyze efficient learning of general exponential families; Empirical exploration is an interesting future direction.**
>
> We thank reviewer sAxR for their detailed comments and feedback.
>
> We respond to the main review below:
>
> We would like to emphasize that we have focused on the theoretical analysis of efficient learning of general exponential families in this work. To that end, we would like to highlight certain distinguishing contributions of our work -
>
> 1. To be the best of our knowledge (and as pointed out by reviewer kDGo), our work is the _first_ to focus learning of general exponential families in a computationally and statistically efficient manner.
>
> 2. We provide rigorous theoretical guarantees (as pointed out by Reviewer SSuU) on the sample complexity and the computational complexity of our algorithm.
>
> 3. We propose a novel computationally efficient estimator and our algorithm (Algorithm 1) is practical. More specifically, the optimization in (11) is a convex minimization problem (i.e., minimizing a convex function over a convex set) and various off-the-shelf efficient implementations of projected gradient descent algorithm can be used.
>
> 4. In addition to consistency and asymptotic normality of our estimator, we also provide an intuition for our approach via Theorem 4.1 and its proof.
>
> That being said, we do agree with the reviewer that an empirical study of the proposed method will help understand its practicality and contrast it with previous works. Building on our current work, we plan to conduct a detailed empirical exploration in a subsequent work.

---

> > ### Comment · Reviewer_sAxR · 2021-08-19
> > **Experimental study**
> >
> > I would like to thank the authors for their responses.
> >
> > I still think that an experimental study is important to convince readers of the significance of the proposed estimator. However, after the reconsideration based on the responses from authors and the comments from the other reviewers, I understand that the paper provides enough theoretical results about the finite sample complexity and computational complexity of the proposed estimator and that these results are good enough to make up for the lack of an experimental study. Based on this observation, I increased my score to 6.

---

> > > ### Author Response · Authors · 2021-08-31
> > > **Thanks!**
> > >
> > > We thank the reviewer for taking out the time to go over our responses! We also thank the reviewer once again for their valuable feedback/suggestions!

---

### Official Review · Reviewer_SSuU · 2021-07-22

**Rating:** 7
**Confidence:** 4

**Summary:**

   In this paper, the authors consider the problem of estimating the
  natural parameters of an exponential family with bounded values from samples.
  With bounded values we refer to the fact that both the norm of the sufficient
  statistics and the norm of the parameters are bounded.

  Their main result is an efficient estimation algorithm with sample
  complexity that is roughly equal to d^2/a^4 where d is the number of
  parameters of the exponential family and a is the estimation error.



**Limitations And Societal Impact:**

Yes

**Main Review:**

  Strengths
  ====================

  - The problem of learning exponential families is very relevant and has a
  wide range of applications in many scientific areas.

  - The proposed solution is elegant and can be applied to real world scenarios
  where a theoretical guarantee is important.

  - The authors do not only provide finite sample results but also asymptotic normality and asymptotic efficiency.


  Weaknesses - Comments
  ========================

  1. The main drawback of the current finite sample result is that it seems
  suboptimal. In particular, in many cases, e.g., Gaussian distributions,
  exponential distributions, linear regression, Poisson distributions, etc., the
  sample complexity for parameter estimation is of order d/a^2 instead of
  d^2/a^4. Is their an example of an exponential family where d^2/a^4 are
  necessary?

  2. Example applications of Theorem 4.3 are missing. What is the result if we
  apply this theorem to bounded value exponential families like Ising models or
  Mallows distributions or truncated Gaussian distributions. For the latter,
  how are the results compared to the recent work of [*].

  3. There is one way to compute the maximum likelihood estimation via sampling.
  In particular, it is not hard to see that if we can sample any given member
  of the exponential family then we can compute an unbiased estimation of the
  gradient of the log-likelihood function. Using this unbiased gradient we can
  run the Stochastic Gradient Descent algorithm which will converge because the
  log-likelihood function is convex. One particular property that is important
  for this is Assumption 4.1 which states that the log-likelihood function is
  not only convex but also strongly convex. Since Assumption 4.1 is necessary
  for the main result of this paper it is important that the lower bounds for
  computing MLE holds even under the Assumption 4.1. Is this true based on the
  known lower bounds?

  4. In a lot of places in the paper the use of the expectation operator is not
  very clear. For example, in Assumption 4.1 it is not clear what is the
  expectation over. My guess is that the expectation is over x ~ f_x(\Theta)
  and the assumption should hold for all \Theta \in \Lambda.

  5. The result cannot be applied to many natural exponential families, e.g.,
  exponential families with unbounded support and sufficient statistics that
  are polynomial functions.

  [*] https://arxiv.org/abs/1809.03986


  Summary of Recommendation
  ======================================
  The results provided in this paper seem to be important towards a better
  understanding of the important problem estimating the parameters of
  exponential families from samples. The solution provided is computationally
  efficient and elegant and for this reason I recommend acceptance.

**Time Spent Reviewing:**

12

---

> ### Author Response · Authors · 2021-08-10
> **Finite sample complexity in-line with prior work; No known lower bounds; Interesting suggestions for future work**
>
> We thank Reviewer SSuU for their detailed feedback and suggestions.
>
> We respond to weaknesses / comments below :
>
> 1. We believe the quadratic dependence of the sample complexity on $d$ is in-line with some of the prior work. For example, the sample complexity of the algorithm for learning truncated Gaussian distributions analyzed in [A] requires a quadratic dependence on $d$. Similarly, the $1/\alpha^4$ dependence in the sample complexity is in-line with some of other prior works (e.g., [40], [51]) that use a similar loss function. It seems that this dependence on $\alpha$ is fundamental to the loss function considered. There is a brief discussion on this in Appendix E.1 of [51]. We do know that for _sparse_ graphical models (e.g., Ising models), one could achieve a better dependence of $1/\alpha^2$ (see [B, C]). However, it is not yet clear what the lower bound on the sample complexity would be for the general class of exponential families considered in this work (which may not be sparse) or how this lower bound depends on $\alpha$. We believe this is an important question and worth pursuing.
>
> [A] ''Efficient Statistics, in High Dimensions, from Truncated Samples'' by Daskalakis et al. (2020)
>
> [B] ''Learning graphical models using multiplicative weights'' by Klivans et al. (2017)
>
> [C] ''Interaction screening: Efficient and sample-optimal learning of ising models'' by Vuffray et al. (2016)
>
> 2. We thank the reviewer for this suggestion. A superficial comparison with the approach proposed in [A] for learning truncated Gaussian distributions leads to a conclusion that the framework developed in this work for general exponential families would suffer by a factor of $1/\alpha^2$ in the sample complexity. However, for a conclusive comparison, a thorough analysis is required. This could be an exciting direction for future research.
>
> [A] ``Efficient Statistics, in High Dimensions, from Truncated Samples'' by Daskalakis et al. (2020)
>
> 3. We agree with the reviewer that the Assumption 4.1 is necessary to show the strong convexity of the loss function. However, it is not clear what known lower bounds the reviewer is referring to. To the best of our knowledge, there are no known lower bounds on learning the class of general exponential families considered in this work. We appreciate further feedback on this.
>
> 4. We thank the reviewer for pointing this out. We will revise the paper to make this clear. As far as the expectation in Assumption 4.1 is concerned, the reviewer is correct --- the expectation is over $\mathbf{x} \sim f_{\mathbf{x}}(x; \Theta^*)$ and the assumption should hold for all $\Theta^* \in \Lambda$.
>
> 5. We acknowledge that the framework developed in this work cannot to applied to exponential families with unbounded support since we assume boundedness of the support. While, conceptually, _most_ non-compact distributions could be truncated by introducing a controlled amount of error, we believe this assumption could be lifted. We describe a couple of ideas in this direction in the limitations section of paper for future work.

---

> > ### Comment · Reviewer_SSuU · 2021-08-31
> > **Thanks for the clarifications**
> >
> > Thanks a lot for clarifying all my questions! I think it would be good to include these clarification in the paper.
> >
> >   Let me be more precise about my question 3: In line 83 of the paper you refer to the results of [23] and [48] as an argument for the computational hardness MLE which makes it non-applicable. Nevertheless, you also make the additional strong assumption Assumption 4.1. My question is:
> >
> >   If we restrict our attention to instances that satisfy the Assumption 4.1 do the hardness results of [23] and [48] still hold? Or, could it be that under Assumption 4.1 MLE becomes computationally tractable?

---

> > > ### Author Response · Authors · 2021-08-31
> > > **Thanks!**
> > >
> > > We thank the reviewer for taking out the time to go over our responses! We will incorporate the reviewer's suggestions in the revised paper.
> > >
> > > Regarding question 3: Thanks for making the question more precise. We believe MLE is computationally intractable even under Assumption 4.1. To see why this is true, it might be useful to focus on parameters that are associated with a _sparse_ graphical model (e.g., Ising models) which is a special case of the setting considered in our work. For such _specific_ parameters, Assumption 4.1 is proven (e.g., Appendix T.1 of [40] provides one such analysis for a condition that is equivalent to Assumption 4.1 for _sparse_ continuous graphical model). However, the MLE associated with a _sparse_ graphical model is still computationally intractable.

---

### Decision · Program_Chairs · 2021-09-28

**Decision:**

Accept (Poster)

**Comment:**

The paper proposes a computationally efficient estimator for learning truncated exponential family distributions from i.i.d. samples. In addition, the proposed estimator is consistent and asymptotically normal.

This paper was viewed as a borderline paper.

There is a well-known tradeoff between computation and statistics. That is, something that is more computationally efficient might be less statistically efficient. In addition, the optimization problem does not use any sparsity-encouraging or low-rank prior. Thus, the polynomial sample complexity is somewhat expected (versus a logarithmic one.)

Unfortunately, it is unclear whether the same arguments would be possible for an optimal method for learning sparse models. (Perhaps from restricted strong convexity but it is ultimately the responsibility of the authors to provide a full proof.) Regarding presentation, I also concur that the tensor notation might be unnecessary. In addition, while not necessary, NeurIPS readers would appreciate synthetic experiments on simple examples, such as truncated Gaussian distributions or discrete pairwise MRF for a small number of nodes.

On the other hand, the paper proposes a new idea that can inspire other researchers in other problems/areas. Specially given that the focus is on such a fundamental problem as finding alternatives to MLE. In particular, the paper proposes a sum of factor potentials (corrected with a uniform distribution), instead of MLE which uses the sum of log factor potentials and the (NP-hard) log partition function.

**Consistency Experiment:**

NeurIPS has a long history of experimentation. In 2014, NeurIPS ran an experiment in which 10% of submissions were reviewed by two independent committees to quantify the randomness in the review process. This year, we repeated a variant of this experiment to see how the quality of the review process has changed over time.  This paper was part of the experiment and was therefore assigned to two committees (consisting of reviewers, an Area Chair, and a Senior Area Chair) that reached independent decisions.  If both committees made the same recommendation, this recommendation was followed. If a single committee recommended acceptance, the paper was accepted (with the exception of a few cases in which the other committee identified what we considered a fatal flaw, e.g., an error in a key result).

Both committees reached the same decision: **Accept (Poster)**

The other committee assigned to the paper recommended **Accept (Poster)**.  You can find the other set of reviews, along with any follow up discussion with the authors here:
https://openreview.net/forum?id=fxGT4XaLkpX